



# A map of global peatland extent created using machine learning (Peat-ML)

Joe R. Melton[1], Ed Chan[2], Koreen Millard[3], Matthew Fortier[1], R. Scott Winton[4], Javier M. Martín-López[5], Hinsby Cadillo-Quiroz[6], Darren Kidd[7], and Louis V. Verchot[5]

[1]Climate Research Division, Environment and Climate Change Canada, Victoria, B.C., Canada
[2]Climate Research Division, Environment and Climate Change Canada, Toronto, ON, Canada
[3]Geography and Environmental Studies, Carleton University, Ottawa, ON, Canada
[4]Institute of Biogeochemistry and Pollutant Dynamics, ETH Zurich, 8092 Zurich,Switzerland. Department of Surface Waters, Eawag, Swiss Federal Institution of Aquatic Science and Technology, 6047 Kastanienbaum, Switzerland
[5]Agroecosystems and Sustainable Landscapes Program, Alliance Bioversity-CIAT, Cali, Colombia
[6]School of Life Sciences, Arizona State University, Tempe, AZ 85287, USA.
[7]Natural Values Science Services, Department of Natural Resources and Environment, Tasmania, Australia

**Correspondence:** Joe R. Melton (joe.melton@ec.gc.ca)

**Abstract.**

Peatlands store large amounts of soil carbon and freshwater, constituting an important component of the global carbon and hydrologic cycles. Accurate information on the global extent and distribution of peatlands is presently lacking but is needed by Earth System Models (ESMs) to simulate the effects of climate change on the global carbon and hydrologic balance. Here,

we present Peat-ML, a spatially continuous global map of peatland fractional coverage generated using machine learning techniques suitable for use as a prescribed geophysical field in an ESM. Inputs to our statistical model follow drivers of peatland formation and include spatially distributed climate, geomorphological and soil data, along with remotely-sensed vegetation indices. Available maps of peatland fractional coverage for 14 relatively extensive regions were used along with mapped ecoregions of non-peatland areas to train the statistical model. In addition to qualitative comparisons to other maps in the literature, we estimated model error in two ways. The first estimate used the training data in a blocked leave-one-out

cross-validation strategy designed to minimize the influence of spatial autocorrelation. That approach yielded an average $r^2$ of 0.73 with a root mean squared error and mean bias error of 9.11% and -0.36%, respectively. Our second error estimate was generated by comparing Peat-ML against a high-quality, extensively ground-truthed map generated by Ducks Unlimited Canada for the Canadian Boreal Plains region. This comparison suggests our map to be of comparable quality to mapping products generated through more traditional approaches, at least for boreal peatlands.



## 1  Introduction

Peatlands are estimated to cover about three percent of the land surface, but contain approximately one third of the soil carbon, and roughly one tenth of surface freshwater (Joosten and Clarke, 2002; Jackson et al., 2017), and are vulnerable to destabilization due to climate change and anthropogenic pressures including drainage and land use change. Their importance in the carbon and hydrologic cycles motivates their inclusion in Earth system models (ESMs) to better understand their potential

impact on the climate system. Since the land surface of ESMs is grid-based, a prerequisite for integrating peatlands into these models is to define the location and the fractional cover of peatlands on the model grid. However, peatlands have generally been overlooked in landscape databases and their mapping remains challenging (e.g. Krankina et al., 2008; Minasny et al., 2019).

As peatlands are commonly considered a type of wetland that contains large amounts of organic carbon in the soil, several

studies have set peatland distribution based on maps of soil organic matter density (e.g. Wania et al., 2009; Bechtold et al., 2019; Hugelius et al., 2020). However, using soil organic matter databases alone in determining peatland distribution tends to overlook the vegetation and subsurface hydrology, but most importantly they rely heavily on the fidelity of the soil carbon dataset. Another approach has been to use a soil map together with global wetland maps or inundation extent maps (e.g. Köchy et al., 2015). These wetland and inundated area databases have mostly been produced through mapping of shallow surface

water based on remote sensing data, as in the Global Inundation Extent from Multi-Satellites initiative (GIEMS; Prigent et al., 2007; Papa et al., 2010) and the Surface WAter Microwave Product Series (SWAMPS; Schroeder et al., 2015) or land cover mapping using surface observations and moderate resolution imaging spectroradiometer (MODIS) data as in the Global Lake and Wetlands Database (GLWD-3; Lehner and Döll, 2004). These wetland mapping products are, however, of limited utility for peatland modelling applications as they generally do not agree well amongst themselves (Melton et al., 2013), which is also

the case for peatland mapping products as is discussed later, and may exhibit biases depending on how they were generated (see discussion in Bohn et al., 2015). As well, in the boreal zone and some areas of the tropics such as the Amazon (Lähteenoja and Roucoux, 2010), some peatlands are not inundated, therefore using hydrological characteristics alone can underestimate their extent (Matthews, 1989; Prigent et al., 2007). Other studies, such as Largeron et al. (2018) or Leifeld and Menichetti (2018), have used a global peatland distribution map derived from a paleontological perspective (Yu et al., 2010). However,

Yu et al. (2010) is an estimated map of binary polygons that does not provide quantitative information on fractional coverage. The most comprehensive global peatland map we are aware of is PEATMAP (Xu et al., 2018), which was generated through a meta-analysis of regional-scale mapping products of varying spatial resolution and provenance (general land cover maps, soil databases, and a hybrid expert system). This dataset is not well suited as a peatland mask for ESM use as the resolution of some





of its parent datasets leaves large polygons of complete peatland cover in regions where this is unlikely and it misses peatlands
in regions where peatland coverage is known to exist, e.g. the Russian Republic of Sakha (Yakutia), as it is dependent upon
mapping products existing for each region.

In describing their dataset, Yu et al. (2010) state that, "accurate true peatland coverage and distribution is not available for
many mapped regions". Over a decade after the publication of Yu et al. (2010), this statement remains accurate. Peatlands have
traditionally been mapped through field surveys and manual inspection of aerial photography (e.g., Tarnocai et al., 2011). These
approaches are costly and labour-intensive and become impractical as the study region becomes large or remote. As noted by
Loisel et al. (2017), it is also difficult to distinguish upland forests from forested peatlands in the boreal region, and between
(sub)arctic tundra vegetation and peatlands in the higher latitudes using aerial photography. Digital soil mapping (DSM) is
an alternative approach to determining global peatland cover. DSM techniques typically combine field surveys with peatland
covariates and statistical models to produce maps of predicted peatland area (McBratney et al., 2003). Following Minasny et al.
(2019), the peatland covariates useful to DSM can be determined from the drivers of peatland formation, indicators of peat
presence, and sensors able to measure the indicators.

The drivers of peatland formation are scale-dependent (Limpens et al., 2008) and thus the intended spatial extent and map-
ping resolution of the DSM product is an important consideration. For DSM on a regional to global scale, as is the case when
mapping for ESM use, the principle drivers of peatland formation are climate, vegetation, and terrain. Minasny et al. (2019)
suggest, for these drivers at this spatial scale, the indicators of peatland presence are climate data (primarily temperature and
precipitation), land use and land cover information, and elevation, slope and terrain attributes. Possible sensors for regional
to global scale mapping include optical and radar imagery, topographic remote-sensing data (digital elevation models; DEMs)
along with climate datasets. The statistical models used as part of DSM vary but here we use a machine learning (ML) al-
gorithm to derive a global map of peatland extent intended for use in ESM applications. As field surveys are impractical to
conduct on a global scale, we rely upon peatland mapping studies on regional scales to train our ML models and evaluate their
results. In Section 2 we define peatlands in the context of our mapping approach as well as describe the datasets used for model
training and the ML approach and algorithms used. Section 3 discusses the results of the ML algorithms as well as our model
performance estimation strategy and limitations of our approach. Section 4 presents our overall conclusions.

## 2    Materials and Methods

### 2.1    Definition of peatlands

As there is no single, universally adopted, definition of peatlands, we follow Joosten and Clarke (2002) in defining them as
areas with, or without vegetation, that contain a naturally accumulating peat layer at the surface. While the definition of peat,
as defined by the percent dead organic material by dry mass, varies considerably in the literature (e.g., Gumbricht et al., 2017;
Page et al., 2011), we choose a more inclusive lower minimum value of 30% to ensure we can capture the diversity of global
peatlands. When using peatland mapping datasets that contain continuous peat depths (Section 2.3), we have used a minimum



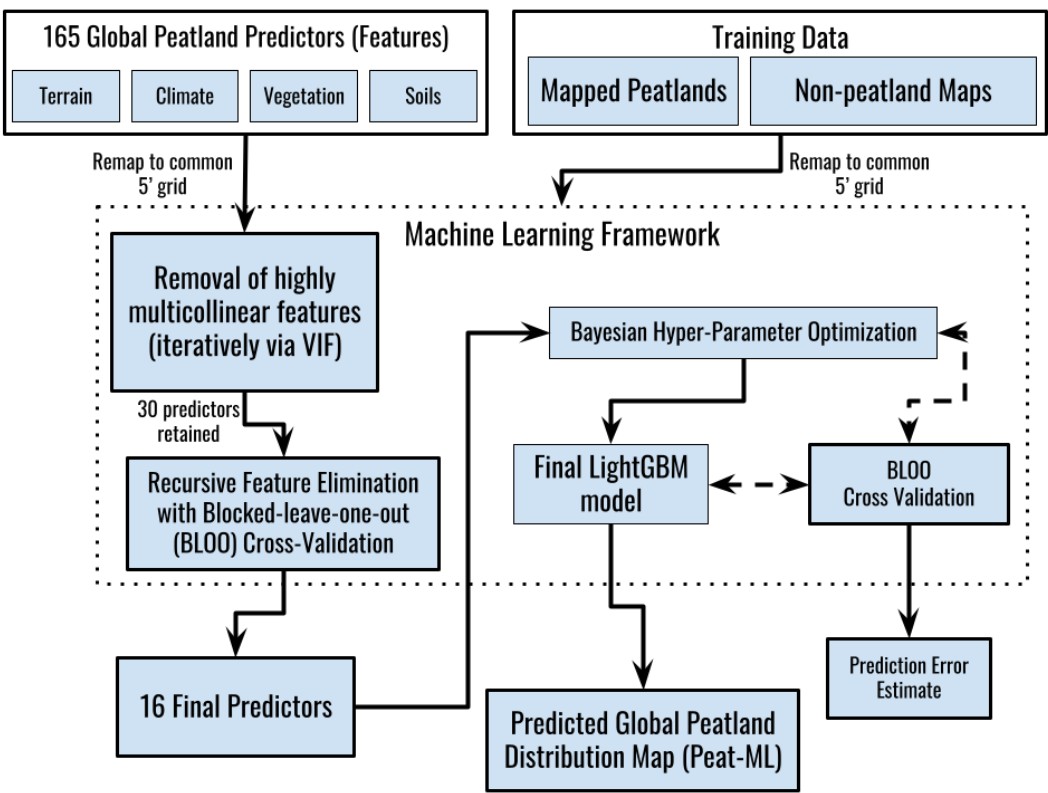

**Figure 1.** Flow chart of the machine learning procedure.

thickness of 30 cm of peat to delineate peatlands, similar to Gumbricht et al. (2017). This depth limit is the most common amongst national datasets (but see discussion on exceptions or the implications of different values in Loisel et al. (2017)).

## 2.2 Data acquisition and preparation

The general process of data preparation, model training and evaluation is illustrated in Fig. 1. All training (regional peatland
and non-peatland mapping products) and predictor (peatland covariates) data were converted from their native format (commonly GeoTiff rasters or vector-based GIS formats such as shapefiles) to netCDF format and remapped onto a common 5 arc minute (ca. 0.0833°, corresponding to 9.26 km at the equator and 4.63 km at 60°N) grid using Climate Data Operators (CDO; Schulzweida, 2020), Geospatial Data Abstraction software Library (GDAL/OGR; GDAL/OGR contributors, 2021), and/or NCO (Zender, 2008). The original resolutions of the data sources are each listed below. All ML runs and evaluation
were performed on the 5 arc minute grid.



### 2.2.1 Predictors (peatland covariates)

We used a suite of predictors that fell into four main types: climate, soils, vegetation, and terrain (geomorphology). Table 1 lists each predictor grouped by predictor source and type. The climate, vegetation, and soil predictors were extracted from the Google Earth Engine data catalogue (Gorelick et al., 2017). The geomorphological dataset was downloaded directly from
its authors' website (Amatulli et al., 2020, Accessed Jan 16th, 2020). Sampling across the different years provided by each dataset is assumed to be relatively unimportant as peatland extent is not highly dynamic across decadal timescales, especially considering the scale of our gridcells (Loisel et al., 2013). An additional predictor was the calculated length of the longest day of the year (hours) for each cell on the 5 arc minute grid.

The climate predictors were derived from the TerraClimate dataset (Abatzoglou et al., 2018). TerraClimate is available at
high spatial resolution (1/24°) and provides monthly climate and climatic water balance variables spanning the 1958 to 2015 period. TerraClimate uses the WorldClim dataset for high-spatial resolution climatic normals, which is combined with the time varying climate of the Climate Research Unit Ts4.0 (CRU Ts4.0; Harris et al., 2020) where the time-varying anomalies of CRU Ts4.0 are interpolated to the high resolution climatology of WorldClim. The Japanese 55-year Reanalysis (JRA55; Kobayashi et al., 2015) is used to in-fill where CRU Ts4.0 has no climate stations contributing to its record (such as part of South America,
Africa, and smaller islands) and was the sole data source for solar radiation and wind speeds. Abatzoglou et al. (2018) notes that the water balance model, used to generate some of the variables listed in Table 1, is simple and does not account for vegetation heterogeneity nor their physiological response under varying environmental conditions. For the climate predictors, we computed seasonal means across the available years, i.e. December - February (DJF), March - May (MAM), June - August (JJA), and September - November (SON). Given that these seasonal means are likely less important in tropical regions, we
did investigate using annual minimum and maximum values in place of seasonal ones but did not see a significant impact on predicted peatland fractional cover.

Soil predictors were obtained from the 250 m resolution OpenLandMap (Hengl, 2018) including soil bulk density (kg m$^{-3}$), clay content (%), sand content (%), organic carbon content (%), and soil water content at field capacity (33kPa). These soil variables are derived from an ensemble of machine learning algorithms trained on a global compilation of soil profiles (Hengl
and MacMillan, 2019). We used the 30 cm depth estimate for all soil variables.

Terrain information is provided by the 250 m resolution version of Geomorpho90m (Amatulli et al., 2020) for 17 different geomorphometry variables describing numerous aspects of the land surface (see Table 1). This geomorphology dataset has an original resolution of 90 m, the same resolution as the Multi-Error-Removed Improved Terrain (MERIT) DEM (Yamazaki et al., 2017) from which it was derived. MERIT-DEM is a merged and error corrected product based on the ALOS World 3D -
30 m (AW3D30) and Shuttle Radar Topography Mission (SRTM3) datasets.

Information about the vegetation state was provided by several datasets. Shimada et al. (2014) created a seamless global mosaic image from Phased Array type L-band Synthetic Aperture Radar (PALSAR/PALSAR2). This image was created with 25 m grid cells on an annual timescale. In creating the mosaic, at each location within a year, the images chosen were those showing minimum response to surface moisture. The images were then ortho-rectified, slope-corrected, and had a destriping





procedure equalize differences between strips that could occur due to conditions at time of acquisition. As the dataset's intended
purpose was to provide a global mask of forest cover (Shimada et al., 2014) soil moisture differences were purposefully
minimized, and thus this dataset is likely of more limited use to predict peatland extent than would otherwise be expected
for a L-band radar product (Izumi et al., 2019; Touzi et al., 2018). However, likely due to the significant computational effort
required to produce a global L-band product, we are not aware of another product publically available.

The MODIS Terra net primary productivity product (MOD17A3.055 NPP) is available annually on a 1 km grid (Running
et al., 2011). This version of MODIS NPP (v. 5.5) is corrected for issues relating to cloud-contaminated MODIS leaf area index
- fraction of photosynthetically active radiation (LAI-FPAR) inputs to the MOD17 algorithm. We used the available 2000 –
2015 period.

   Vegetation indices are provided by the Suomi National Polar-Orbiting Partnership (S-NPP) NASA Visible Infrared Imaging

Radiometer Suite (VIIRS) product VNP13A1 which is generated by selecting the best pixel at 500 m resolution over a 16-day
acquisition period. The VIIRS data is generated for three vegetation indices including the Normalized Difference Vegetation
Index (NDVI), which uses both red and near-infrared (NIR) bands, and two Enhanced Vegetation Indexes (EVI, EVI2) which
also include the blue band with EVI2 designed for intercomparison with other EVI products that don't use a blue band (Table
1). EVI is more sensitive to canopy cover while NDVI is more sensitive to chlorophyll (Huete et al., 2002). All snow, cloud, or

cloud shadow pixels and any pixels that were not excellent, good or acceptable quality (according to the dataset's quality flags)
were excluded. Given the original data does not have composite monthly values, the mean, minimum, maximum, and standard
deviation were all calculated based upon all values within a year then the average was taken across all years.

   Vegetation phenology information is provided by the MCD12Q2 V6 Land Cover Dynamics product (Friedl et al., 2019).
The MODIS vegetation phenology product provides phenological information such as the dates of green-up, peak, and senes-

cence along with variables related to the range and summation of the EVI (see Table 1). Since this is an annual product the
mean/min/max/stdev values are calculated across all years.

   We also considered the Global Surface Water (GSW) dataset of Pekel et al. (2016) but did not include it as a predictor
dataset. We found this dataset to be unsuitable for peatland prediction due to its reliance on Landsat imagery. Treed peatlands,
peatlands smaller than 30 m by 30 m, as well as peatlands where the water table is below the peat surface, such as bogs, would

not be well captured by GSW. A visual inspection of GSW over some of our training regions (see Section 2.3) showed poor
correlation between GSW water presence and mapped peatland area.

## 2.3 Training data

For training and testing the machine learning model, peatland fractional cover was selected as the target variable. However,

accurate estimates of peatland fractional cover are not widely available as discussed in the Introduction. Recently, Minasny
et al. (2019) reviewed the present state of peatland mapping. They found 90 recent studies mapping peatlands with many
delineating peatland extents using ecological and environmental field studies in combination with land cover from remote
sensing, however the studies seldom conduct validation of their mapping and uncertainty estimates are rare (e.g. Table 4 in





**Table 1.** Potential peatland co-variates used as predictor variables for the ML algorithms to predict peatland fractional cover. The treatment of variables is discussed in Section 2.2.1. The predictor variables in bold were selected for the final model (see Section 2.4.3).

| Type | Source | Predictor |
|---|---|---|
| Climate | TerraClimate (Abatzoglou et al., 2018) | Actual evapotranspiration[1], climate water deficit[1], **soil water**[1], potential evapotranspiration (Penman-Monteith), precipitation accumulated, **downward surface shortwave radiation**, **snow water equivalent**[1], **runoff**[1], **Palmer Drought Severity Index (PDSI)**, minimum temperature, maximum temperature, **vapour pressure**, vapour pressure deficit, **10 m wind speed** |
| Soils | Open Land Maps (Hengl, 2018) | **Soil bulk density**, clay content, sand content, soil water content, at field capacity (33 kPa) **organic carbon content** |
| Terrain | Geomorpho90m (Amatulli et al., 2020) | **Slope**, aspect, eastness, northness, convergence index[2], compound topographic index[3], stream power index[4], first and second directional derivatives (East-West, North-South), profile curvature[5], **tangential curvature**[6], elevation standard deviation, **geomorphology landform**[7], roughness indices, topographic position index, **maximum elevation deviation** |
| Vegetation | PALSAR/PALSAR2 (Shimada et al., 2014) | HH[8] and HV[9] polarization backscattering coefficient |
| | MOD17A3 V055 (Running et al., 2011) | **Net primary productivity** |
| | S-NPP VIIRS Vegetation Indices (VNP13A1) (Didan and Barreto, 2018) | Enhanced Vegetation Index (EVI)[10], EVI2[11], near-infrared radiation (NIR), shortwave infrared radiation reflectance (SWIR1[12]), SWIR2[13], **SWIR3**[14], normalized difference vegetation index (NDVI), NIR reflectance[15], green reflectance[16], blue reflectance[17], red reflectance[18] |
| | MODIS Global Vegetation Phenology (MCD12Q2 V6 Land Cover Dynamics) (Friedl et al., 2019) | Dormancy, EVI_Amplitude, EVI_Area, EVI_Minimum, Greenup, Maturity, MidGreendown, MidGreenup, Peak, **Senescence** |
| Geographic | Calculated | Length of the longest day of the year in hours |

[1] derived using a one-dimensional soil water balance model  [2] Ranges between 100 for sinks (convergent areas) and -100 for ridges (divergent areas). Flat areas are 0.  [3] also known as topographic wetness index (Beven and Kirkby, 1979)  [4] Defined as the product of the tangent of the local slope angle and the upstream catchment area.  [5] Measures the rate of change of a slope along a flow line. Convex slopes accelerate water flowing along them while concave slopes decelerate flow.  [6] Measures the perpendicular rate of change to the slope gradient. This captures the convergence (concave curvature) and divergence (convex curvature) of flow across a surface  [7] e.g. flat, spur, valley, etc., calculated using morphometry techniques based on pattern recognition  [8] horizontal transmit and horizontal receive  [9] horizontal transmit and vertical receive  [10] 3-band enhanced vegetation index  [11] 2-band EVI using only red and NIR band  [12] 1230 - 1250 nm  [13] 1580 - 1640 nm  [14] 2225 - 2275 nm  [15] 846 - 885 nm  [16] 545 - 656 nm  [17] 478 - 498 nm  [18] 600 - 680 nm

Minasny et al., 2019). Additionally, the definition of peat can vary between countries and studies (e.g. Table 2 in Minasny

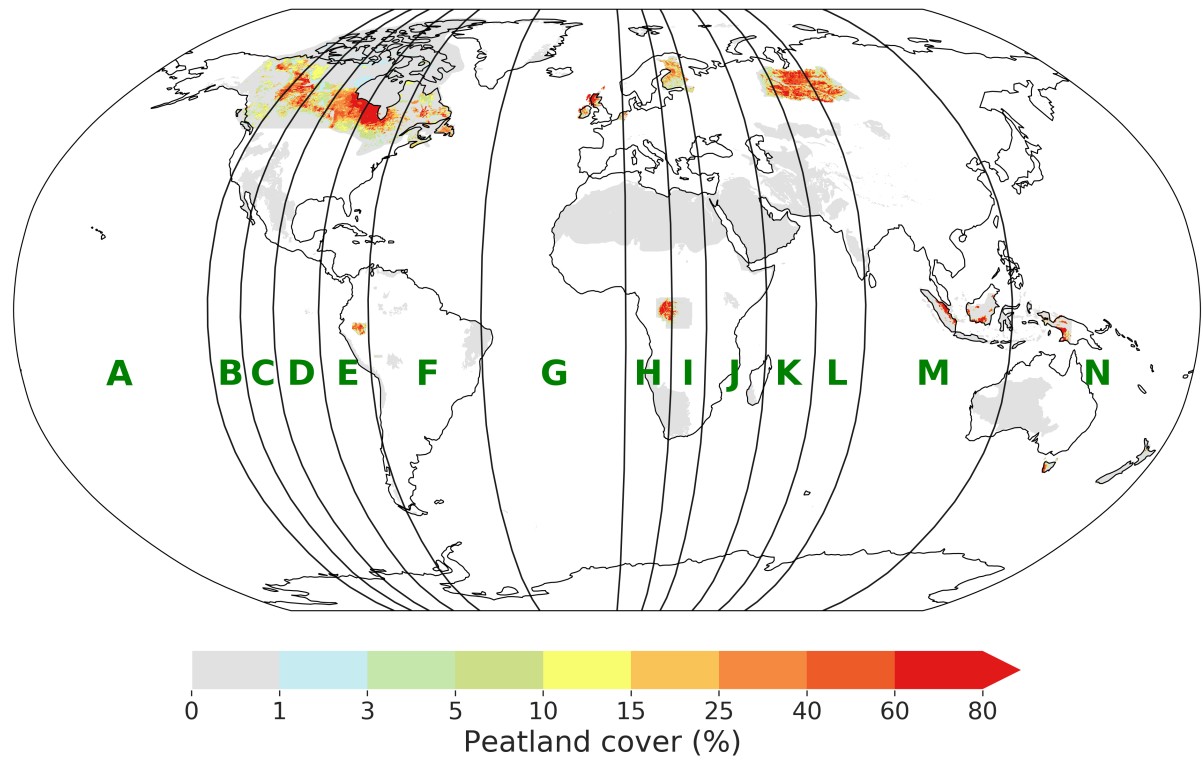

**Figure 2.** Training data for the LightGBM algorithm. Areas in white have no data. The green letters denote the blocks used for the cross validation scheme. The training block limits were chosen as described in Section 2.4.2.

et al., 2019) making assembling an internally-consistent global dataset of peatland extents challenging. In selecting peatland extent estimates for our training data we have chosen studies that are of sufficiently large spatial extent (tens of thousands of square kilometers, but we allow smaller mapping products if they are located in highly under-represented regions), have attempted to validate their peatland extents, and are readily available in digital formats. We have acquired peatland extent estimates for 14 major regions (Fig. 2) including Canada, the taiga zone of the West Siberian Lowlands (WSL), Scotland, the
Netherlands, the St. Petersburg region of Russia, New Zealand, Tasmania, the Cuvette Centrale in the Congo, Indonesia, the Pastaza-Marañón foreland basin (PMFB) in Northwest Peru, and the peatlands along the Peruvian Rio Madre de Dios, along with some peatland-free regions.

Peatland coverage data for Canada, which has ca. 13% of the land surface covered with peatlands, comes from Ducks Unlimited Canada (hereafter DUC; Smith et al., 2007) and The Peatlands of Canada database (Tarnocai et al., 2011). Both
datasets defined peatlands as wetlands (bogs, fens, swamps, or marshes) with massive deposits of peat at least 40 cm thick, as is the convention in Canada. The Peatlands of Canada database was primarily derived from soil surveys and air photo interpretation. Shapefiles were available with information on bog, fen and bog/fen features with ≥1% peat coverage (Tarnocai


et al., 2011). The DUC dataset covers the $74.1 \times 10^4$ km$^2$ Boreal Plains region and was derived from a satellite-based remote sensing classification system validated by 5034 field sites (Smith et al., 2007).

The peatlands of the taiga zone of the West Siberian Lowlands (WSL) is estimated by Terentieva et al. (2016) to be 52.4 $\times 10^4$ km$^2$, or 4 – 12 % of the global wetland area. To conduct this mapping, Terentieva and co-workers used a supervised classification scheme for Landsat imagery that was trained on field data and high-resolution images from 28 test sites. They estimate their accuracy at 79% based on 1082 10 x 10 pixel size validation polygons.

     The St. Petersburg region of Russia was mapped by Pflugmacher et al. (2007) using MODIS Nadir bidirectional reflectance 180   distribution function Adjusted Reflectance (NBAR). The MODIS-NBAR reflectances were combined with empirical regression models to determine sub-pixel peatland coverage. To fit the models, Pflugmacher et al. (2007) drew upon forest inventory data for observed peatland fractional cover over 1105 MODIS pixels with half used for model fitting and half for validation. Error analysis showed good prediction capability with correlation with observations of $r = 0.92$ for unmined peatlands. Pflugmacher et al. (2007) found the region to have approximately 10% peatland cover.

The Finnish Geologic Survey superficial deposits 1:200,000 map displays peat deposits at 0 - 30 cm, 30 cm - 60 cm, and >60 cm depth (Geological Survey of Finland, 2018). The dataset was created through air photo interpretation and field mapping with the smallest polygon size about six hectares.

     A database for the peatlands of Scotland was recently published by Aitkenhead and Coull (2019). Peatland cover was determined using back propagation neural networks trained with peatland survey, climate, topography, Landsat imagery, geologic, 190   and land cover data. Aitkenhead and Coull (2019) reports an r$^2$ of 0.67 for peat depth, which we used to determine peatland fractional cover. Peatlands were assumed to have >30 cm peat and pixels with peat deeper than that were assigned 100% peatland cover and 0% elsewhere.

     The Derived Irish Peat Map version 2 (DIPMv2) (Connolly and Holden, 2009) was compiled from the land cover and soil maps of Ireland using a rules-based decision tree methodology. Connolly and Holden (2009) estimate the overall accuracy of 195   DIPMv2 to be 85%. From the DIPMv2, we included raised bogs, low level Atlantic blanket bogs, and high level montane blanket bogs in producing a peatland cover map.

     Wageningen Environmental Research recently updated the Soil Map of the Netherlands (1:50 000 scale) including peat depth using a combination of boreholes and ordinary kriging (Brouwer et al., 2018; Brouwer and Walvoort, 2019). For each region, a number of boreholes were not used in calibration of the kriging model (roughly 10%) and retained for evaluation. Based 200   on evaluation against the validation borehole subset, the average peat depth error varied between regions but was commonly between 10 and 20 cm. We used the peat depth to delineate peatland area based on a threshold of 30 cm where thicknesses greater than that were assumed to be 100% peatland and 0% elsewhere.

     Draper et al. (2014) mapped peatlands for a region of Amazonia in Northwest Peru (the Pastaza-Marañón foreland basin; PMFB). A support vector machine (SVM) classifier was trained with Landsat, ALOS/PALSAR, and Shuttle Radar Topography 205   Mission (SRTM) elevation data. Along with forest census plots and peat thickness measurements, a supervised classification method was used to train the SVM and determine the distribution of peatland vegetation types as well as above and below ground carbon stocks. The three peat forming vegetation types were pole forest, palm swamp and open peatlands.





The Cuvette Centrale is located in the central Congo basin. Dargie et al. (2017) used a digital elevation model (DEM) to remove steep slopes and high ground, optical data (Landsat Enhanced Thematic Mapper, ETM+) to identify probable swamp

vegetation, which we used as a proxy for peatland fractional coverage, and radar backscatter (L-band synthetic aperture radar; ALOS PALSAR) to identify surface water under forest cover. Together these approaches were used to produce a maximum likelihood tree. They then conducted nine transects of length 2.5 to 20 km to ground truth. Most peatlands in this region are located within large interfluvial basins, are largely rain-fed and ombrotrophic. The areal extent of peat in the Cuvette Centrale was estimated to be $14.6 \times 10^4$ km$^2$ (Dargie et al., 2017).

Indonesian peatlands have been mapped by Wetlands International in a series of publications (Wetlands International, 2003, 2004, 2006). The maps have been derived from regional scale maps and project reports, soil maps, Landsat imagery, along with ground truthing. This dataset uses a 30 cm threshold of peat thickness to delineate peatlands.

National maps of New Zealand peatlands were derived from the Fundamental Soil Layers (FSL) soil maps published at 1:50 000 scale by the New Zealand Land Resource Inventory (NZLRI; Landcare Research NZ Ltd, 2000). The polygons in the

FSL maps were manually created from aerial photograph analysis with ground truthing. Peatlands were selected by choosing the organic soils class.

Organic soil and peat mapping was undertaken by the Department of Natural Resources and Environment, Tasmania, to provide decision support for fire management and suppression activities in the Tasmanian Wilderness World Heritage Area (Kidd et al., 2021). A DSM approach was used to predict organic soil and peat areas using new and existing soil site data,

intersected with a range of environmental predictor datasets, which included vegetation mapping, legacy soil mapping, wetlands, digital elevation models and terrain derivatives, remote-sensing (multispectral green/ bare areas, gamma radiometrics, Sentinel RADAR), and climate. A binary 'presence-absence' calibration set of site data was used to create a digital map index (0 – 1). Modelling was undertaken using Regression-Trees with 10-fold cross validation, where spatial output values closer to '1' were deemed to be meeting the environmental conditions conducive to peat formation. The organic soil extent modelling

$R^2$ calibration and validation values were 0.77 and 0.70 respectively. Map validation by expert review determined that spatial index values > 0.75 were highly likely to be peat (or organic) soils (Kidd et al., 2021).

Peatlands along the Rio Madre de Dios in Peru were mapped by Householder et al. (2012) using Landsat imagery and field observations. They identified 295 peatlands from remote-sensing imagery covering 294 km$^2$ and from 0.1 to 35.0 km$^2$ in size. Field verification was performed at 35 peatlands giving over 800 georeference validation data points.

To increase the number of cells for model training and also improve representation of peatland-free landscapes, we included polygons of ecoregions that should contain little to no peatlands from Olson et al. (2001), thus all areas in these ecoregions/biomes were considered to have zero peatland extent. The ecoregions chosen were the global distribution of the Desert and Xeric Shrublands biome, but excluding 15 ecoregions that had a non-zero peatland extent within at least one grid cell according to PEATMAP. This was to ensure we take a conservative approach to the use of these non-peatland masks. Two

South American ecoregions (Beni Savanna and the Rio Negro campinarana; Fig. 2) were also included as peat-free regions. We discuss the inclusion of these ecoregions in Section 3.3. A further region of zero peatland extent was defined according to a map of soil organic carbon for the Casanare flooded savannas of Colombia (Martín-López et al., 2021) and expert opinion





**Table 2.** Training data (regional peatland mapping products) for the machine learning model.

| Region | Source | Peatland determination technique |
|---|---|---|
| Boreal Plains of Canada | Ducks Unlimited Canada (Smith et al., 2007) | Satellite imagery with >5000 site-visits |
| Rest of Canada | Tarnocai et al. (2011) | Primarily from soil surveys and air photo interpretation |
| West Siberian Lowlands (Taiga zone) | Terentieva et al. (2016) | Supervised classification of Landsat trained on field data |
| St. Petersburg region (Russia) | Pflugmacher et al. (2007) | Regression models from MODIS-NBAR reflectance |
| Finland | Geological Survey of Finland (2018) | Field mapping and air photo interpretation |
| Scotland | Aitkenhead and Coull (2019) | Neural networks trained with survey data and covariates |
| Ireland | Connolly and Holden (2009) | Rules based decision tree with land cover and soil maps |
| Netherlands | Brouwer et al. (2018) Brouwer and Walvoort (2019) | Ordinary kriging with boreholes for calibration and evaluation |
| Amazonia[1] | Draper et al. (2014) | SVM supervised classification using elevation, optical and radar remote-sensing data |
| Congo basin (Cuvette Centrale) | Dargie et al. (2017) | Combination of DEM, Landsat ETM+, and ALOS PALSAR along with ground truthing transects |
| Indonesia | Wetlands International (2003,2004,2006) | Collation of regional maps, soil surveys, Landsat imagery verified by ground truthing |
| New Zealand | Landcare Research NZ Ltd (2000) | Collation of regional maps and soil surveys |
| Tasmania | Kidd et al. (2021) | ML with terrain, vegetation mapping and satellite spectra covariates including seasonal Sentinel-1 coverage |
| Rio Madre de Dios (Peru) | Householder et al. (2012) | Landsat imagery with field mapping |

[1] Pastaza-Marañón foreland basin (PMFB) in Northwest Peru

based upon field observations. We also set peatland area to zero for any pixels that are ice covered as shown in the Global Land Ice Measurements from Space (GLIMS) dataset (GLIMS and NSIDC, 2018).


## 2.4 Machine learning approach

### 2.4.1 LightGBM and hyperparameter optimization

The statistical modelling was conducted in the Python programming language (v. 3.8.3). We use a Gradient Boosting Decision Tree (GBDT) algorithm called LightGBM (Ke et al., 2017). Decision tree algorithms make iterative splits to partition data according to different criteria. The decision tree will split each node at the feature with the largest information gain, i.e. the most informative. For GBDTs, the information gain is usually measured by the variance after splitting. To avoid issues with





overfitting of a decision tree, GBDT algorithms use the boosting technique which combines multiple decision trees in series to achieve better predictive power as each tree in the series attempts to minimize the errors in the previous tree. The error minimization steps occur through a form of gradient descent in function space where each tree is trained on a residual vector
that measures the magnitude and direction of the true target relative to the previous tree (loss function), which successive iterations minimize.

### 2.4.2   Cross-validation approach

To provide estimates of the error associated with the LightGBM predictions we adopted a blocked-leave-one-out (BLOO) strategy, which is recommended for applications where the predictors could be expected to exhibit spatial autocorrelation
(Roberts et al., 2017; Meyer et al., 2019; Ploton et al., 2020). BLOO tends to produce estimates of prediction error that are closer to the 'true' error (Roberts et al., 2017), particularly in cases where the sampling strategy is clustered (Rocha et al., 2021). We chose to block our cross-validation (CV) regions based on longitudinal limits to allow both boreal and tropical peatlands to potentially be represented in each block. The optimal number of training blocks is an important determination. Choosing blocks that are too small risks incorrectly increasing our CV-determined model accuracy due to spatial autocorrelation issues, while
choosing overly large blocks will result in information loss and worsens our assessed model accuracy unduly. We determine the optimal number of blocks by comparing the length scale of autocorrelation of the model residuals with our block sizes. Fig. A1 shows the autocorrelation tends to zero at a length scale (sill) of around 500 km. To accommodate this we set a minimum block size of 10° of longitude (which corresponds to roughly 500 km at 65° latitude). Based on the constraints of our minimum block size and the need for a roughly even number of training cells in each block, we end up partitioning the globe into 14
blocks as shown in Fig. 2. The CV was performed by holding out one block, training the LightGBM algorithm over the other blocks and then using that trained model to predict the peatland extent over the held-out block. This was performed for each block in turn and the results averaged to give an estimate of the prediction error.

### 2.4.3   Predictor selection and model optimization

From the potential peatland covariates listed in Table 1, and discussed in Section 2.2.1, we processed 163 global peatland
features that could be used by the machine learning model. However, it is likely that many of these predictors will have low predictive power and duplicate information provided by other predictors leading to over-fitting by the ML algorithm (Dormann et al., 2013). To select only the most relevant features we used both iterative feature removal based on the calculated multicollinearity and Recursive Feature Elimination with Cross-Validation (RFECV)(Pedregosa et al., 2011), which is a form of backward feature elimination.
Multicollinearity was accounted for by using the calculated Variance Inflation Factor (VIF) to identify and remove highly correlated variables (Alin, 2010). VIF uses ordinary least squares regression to determine collinearity with the score determined by

$$\text{VIF} = \frac{1}{(1 - R_j^2)} \tag{1}$$



where $R_j^2$ is the multiple coefficient of determination for the feature $j$ on the other features (covariates). One approach would
be to simply set a threshold VIF value and remove all features with VIF values higher than this threshold in a single step.
However, in order to avoid the elimination of potentially important features, we chose instead to conduct the exclusion process
iteratively. First each feature had its VIF score calculated. Then all features with a VIF value higher than 5 (corresponding to
a $R_j^2$ of 0.8) were ranked according to their information gain calculated by LightGBM and the feature with the lowest gain
was removed. The model was then retrained and the VIF value recalculated. If features remained that had a VIF above the
threshold value, the same ranking and removal would occur until all remaining features had a VIF value below threshold. This
step retained 30 features (listed in Table A1). The VIF value chosen is quite stringent, well below what Dormann et al. (2013)
suggest as a critical value (10).

We use RFECV with BLOO CV (using the same blocks as described in Section 2.4.2) in an iterative manner to ascertain the
optimal number of features. RFCEV first trains the LightGBM algorithm on the original number of features (here 30) with the
features ranked for their importance, based on information gain, on the model's root mean square error (RMSE) as determined
by the BLOO CV. The least important feature is removed and the model is retrained using the new subset of features. By
retraining the model after each feature is held-out, we avoid the issue of extrapolation that can occur in permutation-based
approaches (as described in Hooker et al., 2021). The algorithm can then produce an estimate of model skill as a function
of the number of features trained (Fig. A2). The RFECV algorithm will choose an optimal number of features based on the
greatest model skill. Based on Fig. A2, 16 features (highlighted in Table 1) were selected as the optimal number to retain for
the optimization process and the final model.

GBDT algorithms tend to require hyperparameter tuning to ensure the model is performing optimally. We employed Bayesian
optimization on 11 LightGBM hyperparameters (Table A2) using the *hyperopt* package (Bergstra et al., 2015) over 500 trials.
In each trial, the final 16 predictors identified in the steps above were used in the LightGBM model to optimize the model's
calculated RMSE based upon the BLOO CV. The optimized parameters were then used to generate the Peat-ML map.

## 3 Results and Discussion

### 3.1 Predictor importance

The top ten predictors based on information gain as determined by the LightGBM algorithm are shown in Fig. 3. Based on the
full LightGBM model runs (Peat-ML hereafter), the most informative feature is the geomorphological landform (e.g. flat, spur,
valley, peak, etc.), which is calculated using morphometry techniques based on pattern recognition (Amatulli et al., 2020). The
next most important predictor is terrain slope defined as the rate of change in elevation along the direction of the water flow
line (Amatulli et al., 2020). The third and fourth most important variables are soil organic carbon at 30 cm depth and shortwave
infrared radiation reflectance at 2225 - 2275 nm (SWIR3). The remaining less important features (≲5%) relate to climate (DJF
snow water equivalent, MAM vapour pressure, DJF shortwave radiation and wind speed, SON runoff, and Terraclimate-derived
DJF soil water).



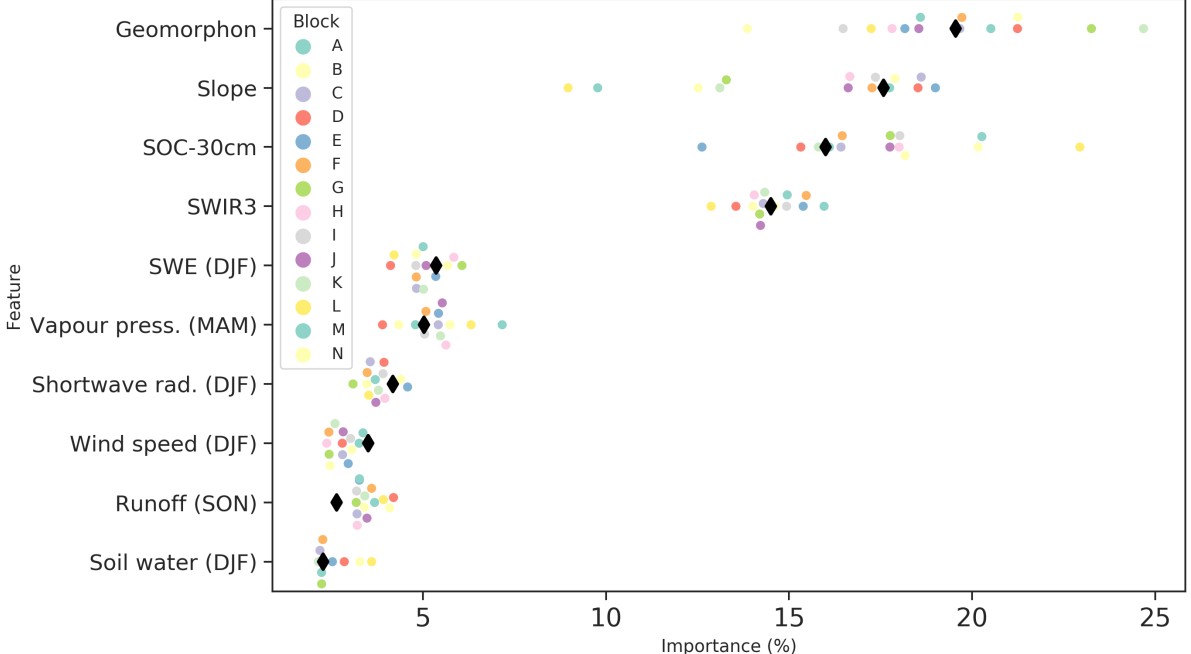

**Figure 3.** Predictor importance based on percent information gain for the top ten features as determined by the LightGBM algorithm. The feature ranking is shown for each of the blocks used during the BLOO CV (coloured dots; see Section 2.4.2). The feature importance from the full model simulation is shown by the black diamonds. SWIR3 is the shortwave infrared radiation reflectance for 2225 - 2275 nm, Geomorphon is the geomorphological landform, SWE is the snow water equivalent, SOC is soil organic carbon at 30 cm depth. See Table 1 and Section 2.2.1 for more details.

Minasny et al. (2019) suggest the indicators of peatland presence on a regional to global scale are climate data (primarily temperature and precipitation), land use and land cover information, and elevation, slope and terrain attributes. Slope has also been used in several terrestrial ecosystem models as a means to predict wetland areas, i.e. the flatter a region, the more likely water will stagnate allowing wetland formation (e.g. Kaplan, 2002; Arora et al., 2018). Interestingly then, the top two

predictors are important components of the Kaplan (2002) wetland determination scheme. The geomorphological features appear to provide further information about the land surface characteristics that can allow peatland formation, distinct from that of slope alone. The importance of the SOC variables demonstrates the close relationship between SOC and peat soils, as has been exploited for peatland mapping in the past (e.g. Hugelius et al., 2020). The importance of SWIR3 likely reflects its utility in determining wet earth from dry earth and providing information about the vegetation water status. SWIR3 is

particularly useful as a feature as it can help delineate fens, as otherwise the ML model lacks a predictor of groundwater contributions to surface water, as well as peatlands from uplands in general as SWIR reflectance is generally sensitive to soil moisture, soil type, and vegetation leaf area index and water content (Wang et al., 2008; Tian and Philpot, 2015). Of the climate predictors, DJF SWE along with DJF shortwave radiation could have been used by the ML model to distinguish boreal from





tropical peatlands. Vapour pressure may also have some utility in determining peatlands due to the differing evapotranspiration
response of peatlands from upland forests (Helbig et al., 2020). In general however, all the climate variables were of relatively
small importance with roughly 5% or less importance as measured by information gain.

Fig. 3 also shows the feature importance as found by the BLOO CV for each block (whereby each block in the figure
shows the feature importance ranking when that block was not trained upon for the CV). Looking at feature importance broken
down in this manner reveals some remarkable consistency in some predictors, e.g. relatively low importance predictors (<10%)
remain consistently less important. While other features have highly variable importance principally slope, geomorphon, and
SOC-30cm. These three variables can switch order of importance when trained to exclude certain training blocks during the
BLOO CV. When trained with all training data (full model; black diamonds in Fig. 3), predictor importance is generally close
to the middle of the range set by the blocks from the BLOO CV, excluding some of the more minor features such as SON runoff
or DJF wind speed. This demonstrates that, given there are only 14 blocks, excluding training data as part of the BLOO CV
can have relatively large consequences, especially as each peatland region has its own particular characteristics as evidenced
by the changing predictor importance. For example, the Cuvette Central, western Amazonia and tropical islands of Asia all
appear to differ significantly for characteristics such as peat depth, structure, carbon density, etc. (see Table 1 in Dargie et al.,
2017).

## 3.2    Predicted peatland extents

### 345    3.2.1    Global

Global peatland extent as predicted by Peat-ML is shown in Fig. 4. When Peat-ML is compared to PEATMAP (Xu et al.,
2018), many major peatlands regions appear similar including Canada, the WSL, the Cuvette Centrale of the Congo, and parts
of Scandinavia. However, the two maps also differ substantially. The regions with the most notable difference between the
two products include Alaska, parts of Africa excluding the Congo, and eastern Siberia. There are more intermediate peatland
extents predicted by Peat-ML whereas PEATMAP tends to show more regions of 100% peatland extent with less gradation
between peatlands. Our estimated global peatland extent at 4.04 $\times 10^6$ km$^2$ is similar to the PEATMAP estimate of 4.23 $\times 10^6$
km$^2$ (Table 3).

Our Northern Hemisphere (>23°N) estimates of 3.0 million km$^2$ is lower than the other available estimates including PEATMAP
(3.2 million km$^2$), the lower bound of Hugelius et al. (2020) (3.2 million km$^2$), and an older estimate of Gorham (1991) but is at
the lower bound suggested by Loisel et al. (2017). In the tropics, our model estimate is roughly the same as PEATMAP, but only
a little over half of the extent estimated by Gumbricht et al. (2017). The Gumbricht et al. (2017) map was produced through a
hybrid approach that uses hydrological modelling, remote-sensing products and hydro-geomorphology from topographic data
along with expert assessment. It is only available across the tropics (maximum 40°N).

The spatial distribution of the predicted peatlands will now be examined in detail. We focus on regions that have either
multiple other peatland mapping products for comparison or contain large areas of predicted peatlands.

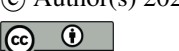

**Figure 4.** Global peatland extent as estimated by Peat-ML along with PEATMAP (Xu et al., 2018).

### 3.2.2 Boreal peatlands: Europe and Russia

Fig. 5 shows the peatland extent in the WSL, western Russia and parts of Scandinavia for Peat-ML, its training data, PEATMAP, Hugelius et al. (2020), and the Boreal-Arctic Wetland and Lake Dataset (BAWLD) (Olefeldt et al., 2021). The Hugelius et al. (2020) dataset is derived from the mean of two soil datasets and is only available for the Northern Hemisphere ($> 23°N$).

The BAWLD product is derived from expert assessment that is then extrapolated through the use of random forest models and





**Table 3.** Peatland extents as estimated by Peat-ML alongside other literature estimates.

| Region | Source | Peatland extent (km$^2$) |
|---|---|---|
| Global | Peat-ML | $4.04 \times 10^6$ |
| | PEATMAP | $4.23 \times 10^6$ |
| Northern hemisphere ($>23°$N) | Peat-ML | $3.00 \times 10^6$ |
| | Gorham (1991)[1] | $3.46 \times 10^6$ |
| | Loisel et al. (2017)[2] | $3.0 - 3.5 \times 10^6$ |
| | PEATMAP | $3.19 \times 10^6$ |
| | Hugelius et al. (2020) | $3.7 \pm 0.5 \times 10^6$ |
| Tropics ($23.5°$S - $23.5°$N) | Peat-ML | $0.96 \times 10^6$ |
| | PEATMAP | $0.94 \times 10^6$ |
| | Gumbricht et al. (2017) | $1.70 \times 10^6$ |
| Canadian Boreal Plains | Peat-ML | $185 \times 10^3$ |
| | DUC | $186 \times 10^3$ |
| | PEATMAP [3] | $185 \times 10^3$ |
| | Hugelius et al. (2020) | $164 \times 10^3$ |
| | Webster et al. (2018) | $269 \times 10^3$ |

[1] Boreal and subarctic peatlands    [2] Suggested best estimate for modern peatland area. Includes a summary of other estimates which range between 2.4 - 4.0 $\times 10^6$ km$^2$.    [3] here PEATMAP's underlying data source is Tarnocai et al. (2011)

geospatial datasets across the boreal and arctic regions. The original spatial resolution is relatively coarse at $1°$ by $1°$. For the WSL region, all four products are similar with only slight differences in the peatland fractional cover, rather than its spatial distribution. Peat-ML shows strong similarity with its training data as would be expected. PEATMAP stands out compared to the other maps due to its almost binary peatland coverage showing either high values or no peatlands with little gradation in

between. Compared to Hugelius et al. (2020), Peat-ML shows less peatlands in the northern edge of the Northwestern region of Russia but more by the White Sea. Both PEATMAP and Peat-ML do not show peatlands near the mouth of the Kara River to the northwest of the terminus of the Ural Mountains as evident in Hugelius et al. (2020) and BAWLD, while Peat-ML and BAWLD show few peatlands on the Yamal Peninsula where both PEATMAP and Hugelius et al. (2020) suggest appreciable extents. Generally, Peat-ML has more similarity to PEATMAP than Hugelius et al. (2020) and BAWLD over the western

Russian domain.

All maps show relatively similar distributions of peatlands surrounding the Baltic Sea (Fig. 5). None of the maps indicate peatlands by the Caspian Sea as seen in PEATMAP except some small extents (1-3% predicted by Peat-ML) to the northwest of those depicted in PEATMAP.

As with Eastern Europe, Western Europe is similar in that PEATMAP shows a more binary representation of the peatland

extent compared to the other maps (Fig. A5). Peat-ML and Hugelius et al. (2020) have fairly similar peatland distributions



**Figure 5.** Maps of eastern European and Russian peatlands including a) training data used by the ML model, b) Peat-ML predicted peatlands along with the peatland coverage from c) PEATMAP (Xu et al., 2018), d) Hugelius et al. (2020), and e) the Boreal-Arctic Wetland and Lake Dataset (BAWLD; Olefeldt et al., 2021).

and extents. The main differences are expressed in small pockets of peatlands, e.g. eastern Spain has scattered peatlands in



Hugelius et al. (2020) that are not found in Peat-ML or PEATMAP, whereas in western Hungary both Hugelius et al. (2020) and PEATMAP show small peatlands not predicted to be as extensive by Peat-ML.

### 3.2.3 Boreal peatlands: Canada and Alaska

The northern USA, Canada, and Alaska peatland extents are shown in Fig. 6. Alaskan peatlands predicted by Peat-ML have some similarity to the Hugelius et al. (2020) map and BAWLD with extensive peatlands in western Alaska (Lower Yukon region). These peatlands are not evident in PEATMAP which shows less extensive, but high coverage, peatlands more along the southern and eastern edges of the state. Peat-ML, Hugelius et al. (2020), and BAWLD predict peatlands along the North Slope, which are not evident with PEATMAP. Other reports suggest extensive wetlands in Alaska (e.g. Glass, 1992) but we are
not aware of any mapping product detailing peatland specific coverage.

For Peat-ML, the Canadian peatlands from Tarnocai et al. (2011) and DUC (Smith et al., 2007) are used as training data, which naturally gives good correspondence between Figs. 6a and 6c. For a more informative comparison of general model skill for boreal peatland regions, Peat-ML predictions from the BLOO CV simulation are also shown as this would give some indication of predictions without the benefit of training upon a particular region's peatlands (Fig. 6b). Generally, all datasets
shown in Fig. 6 display some strong similarities with large peatlands shown for the Hudson's Bay Lowlands (HBL), the Mackenzie Delta, and across the Boreal Plains, yet important differences are also visible. Webster et al. (2018) shows little peatlands along the southern edge of the Hudson's Bay, perhaps due to their peatland determination model's emphasis on treed peatlands. Webster et al. (2018) also show generally higher peatland coverage, where peatlands are present, than the other datasets. Hugelius et al. (2020) predicts extensive but relatively low coverage across much of the Canadian eastern Arctic,
which is not found in any of the other peatland maps. Of the five peatland maps, the most closely corresponding peatland extents appear to be between PEATMAP, BAWLD and Peat-ML.

The northern USA has some peatlands around the Great Lakes evident in PEATMAP and Hugelius et al. (2020) ( 10 - 60%), which are also predicted, but less extensive in Peat-ML (usually  1 - 15%). Besides the coverage differences, the products have a similar spatial extent although PEATMAP's peatlands are more commonly higher coverage per identified peatland.

### 3.2.4 Tropical peatlands: South and Central America

South American peatlands are shown in Fig. 7. Peat-ML peatland training data for this region (Fig. 7a) is currently limited, encompassing only Peru's Pastaza-Marañón foreland basin (PMFB) and the Rio Madre de Dios. In early simulations with Peat-ML, and maps from other modelling processes (e.g., Gumbricht et al., 2017) we noticed predictions for high peatland coverage in areas of South America where peat is not known to occur. This includes seasonally flooded savannas, such as the
Llanos de Moxos (Beni Savanna) and Llanos Orientales of Colombia and Venezuela. A recent field expedition searching for peat in the Colombian Llanos failed to discover any peat deposits (Martín-López et al., 2021), which could indicate that these tropical savanna biomes are generally not able to form extensive peat deposits. Additionally, white sands ecosystems are not known to support extensive peatlands, so we also excluded the Rio Negro campinarana ecoregion that corresponds with white

**Figure 6.** Peatland extents for Canada, northern USA, and Alaska for a) Peat-ML, b) Peat-ML from the BLOO CV, c) the training data used
for the ML model, and three other peatland extent products (d-f).

sandy soils (spodosols/podzols) and not histosols. Without this negative data, we would likely overpredict peat extent in South

America rather severely.

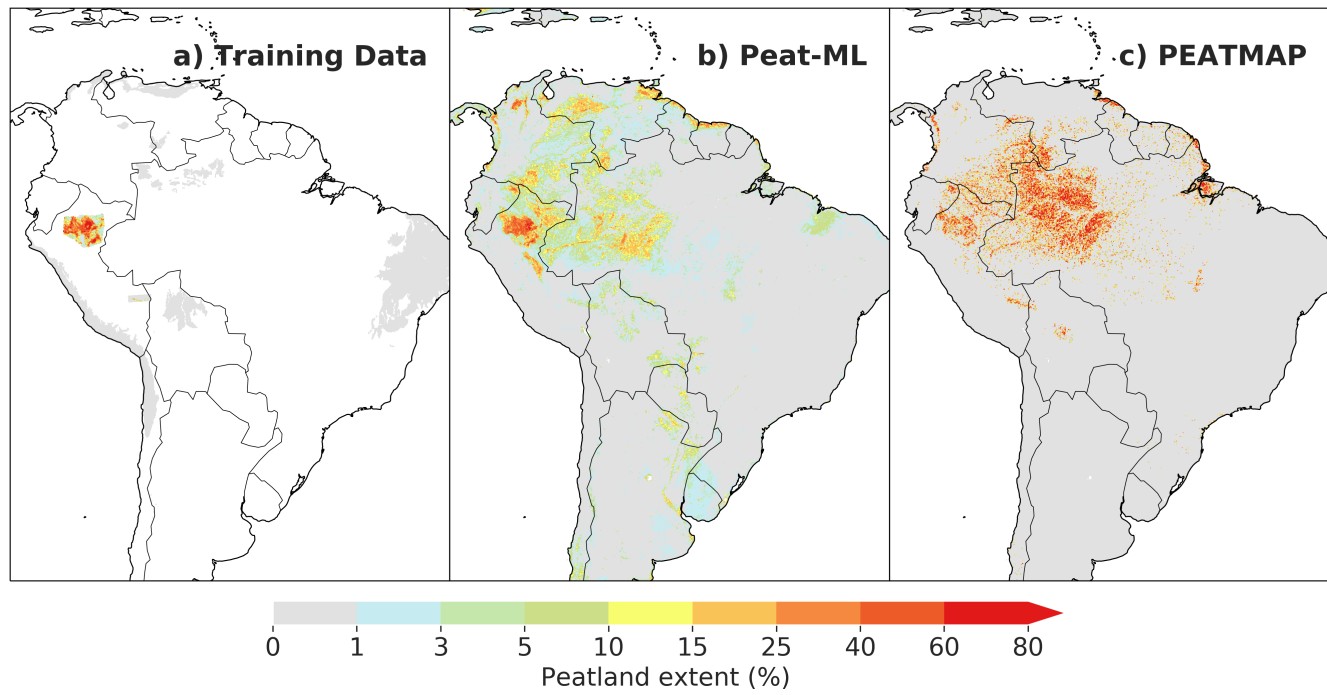

**Figure 7.** South American peatland extents. Panel a) shows Peat-ML training data, b) Peat-ML predicted peatland coverage, and c) PEATMAP (Xu et al., 2018) which is taken from Gumbricht et al. (2017) for this region.

Peat-ML predicts an extensive peatland in the PMFB as well as central Amazonia. The extent of peatlands in this region is less than in PEATMAP, mainly due the generally lower extent per grid cell although in broadly similar regions. Both PEATMAP and Peat-ML show peatlands along the northeastern coast of the continent. Peat-ML predicts smaller peatland extents (generally <10 - 15% coverage) in the Pantanal and along the Paraguay River as it joins the Paraná River down to the

Rio de la Plata, which are not evident in PEATMAP.

There are some non-peatland river floodplains that Peat-ML characterizes as peatlands, such as Colombia's Rio Guaviare. This river may be too dynamic to allow extensive peat formation due to relatively rapid meandering that would scour away peatforming depressions faster than the organic matter can accumulate or else bury potential peats with mineral sediments from the Andes (Junk, 1982). Given the lack of an appropriate predictor for these hydrogeomorphological processes operating on

decadal to century timescales, it is not surprising Peat-ML may overestimate peat extent in these ecosystems. Other areas like Colombia's Amazon catchment region might be susceptible to similar processes as these regions are suggested to be floodplain forests in Ricaurte et al. (2017) however their map is based on the CORINE Land Cover data for Colombia (IDEAM, 2010). Other areas in Colombia where Peat-ML predicts peatlands include parts of the Orinoco catchment region, where Ricaurte et al. (2017) shows flooded grassland savannas and riparian wetlands and the Caribbean catchment region where peatlands

are indicated by CORINE along with other wetland types. Given that the CORINE land cover product is based upon remote-





sensing with little ground truthing, it is possible that several of these wetland regions shown in Ricaurte et al. (2017) are actually peat-forming, making it difficult to definitively evaluate Peat-ML against this dataset. Besides the occasional small peatland area (e.g. in the Páramo of Ecuador; Hribljan et al., 2017), there are few sources of high quality peatland mapping products for South America to evaluate Peat-ML against. While Peru has the PMFB mapped by Draper et al. (2014) and the Rio Madre de
Dios by Householder et al. (2012), and is proposed to have extensive peatlands by Gumbricht et al. (2017) and Peat-ML, there is presently no national peatland inventory (López Gonzales et al., 2020).

Peat-ML predicts more peatland extent than PEATMAP in Central America (Fig. A6). Much of the predicted peatlands are close to coastlines, in particular along the Atlantic coasts of Mexico, Nicaragua, Costa Rica, and Cuba. Peat-ML places more extensive peatlands on the Yucatan Peninsula of Mexico, which is not evident in PEATMAP. A desk-based assessment of
peatlands based upon cartographic approaches with solicited expert assessment shows similar distributions of peatland extent, but with less peatlands in the Yucatan (Peters and Tegetmeyer, 2019). The Yucatan peninsula has relatively extensive marsh and mangrove coastal wetlands but is a karstic landscape with a highly permeable carbonate substrate (Adame et al., 2013) suggesting Peat-ML is overestimating peat extent for the inland portions of the peninsula.

### 3.2.5  Tropical peatlands: Africa and the Indonesian archipelago

African peatlands (Fig. 8) are also poorly mapped making it difficult to evaluate the Peat-ML results. There are notable differences between Peat-ML and PEATMAP. PEATMAP shows very few peatlands outside the Congo's Cuvette Centrale, whereas Peat-ML has relatively extensive peatlands in South Sudan, along the border of the Central African Republic and Chad. This is in general agreement with more qualitative African peatland extent estimates (Grundling and Grootjans, 2016), and demonstrates Peat-ML's ability to reasonably determine peatland extents in regions where reliable spatially-explicit mapping products are absent. Regardless, Peat-ML may still be underestimating African peatlands due to a lack of appropriate training data. An
example is newly documented peatlands in the Okavango delta (Gelinas, 2018), which have a dominantly herbaceous vegetation cover (sedges, papyrus, grasses) while our only training dataset for Africa is a swamp forest (Cuvette Centrale). Future iterations of Peat-ML may profit from some active mapping campaigns presently underway in East Africa (R. Barthalmes, pers. comm. 2021) that could provide much needed training data, and thereby improve predictions, for the peatland regions of Africa. Improving understanding of African peatland extents will likely remain challenging however, due to land use pressures
that may complicate peatland identification and mapping as Grundling and Grootjans (2016) suggest African peatlands are heavily utilized by rural populations that depend on the peatland's water and organic soils for crop cultivation.

While much of the Indonesian archipelago contains training data for the ML algorithm, the neighbouring states of Papua New Guinea, Brunei and Malaysia are entirely model predicted (Fig. A7). While Malaysian peatlands appear similar between Peat-ML and PEATMAP, Papua New Guinea is quite different. PEATMAP shows extensive peatlands in the central mountainous
region of the country, while Peat-ML has the peatlands placed in the surrounding lowland regions. There is some indication that the mountainous regions should have extensive peatlands (Hope, 2015). These peatland complexes appear to be sufficiently different from the Peat-ML training data that the ML model is unable to predict them.



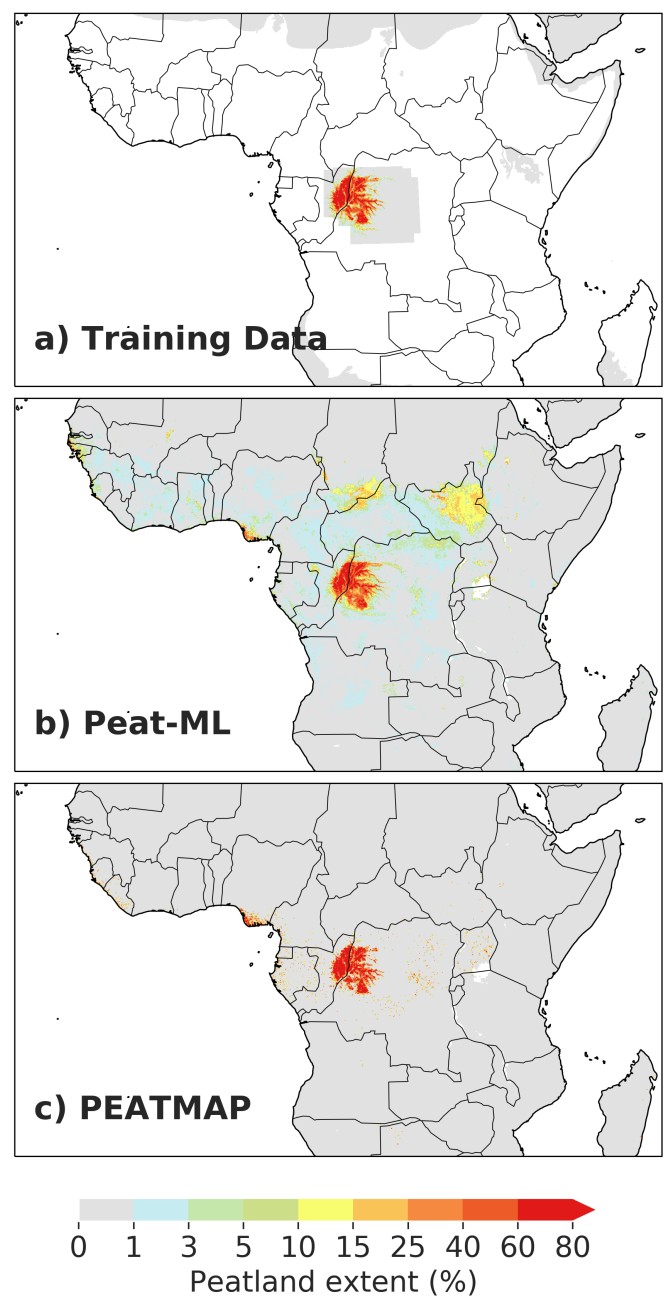

**Figure 8.** Peatland extent over central Africa. Panel a) shows the ML training data, b) the Peat-ML predicted peatland extent, and c) the PEATMAP extent from Xu et al. (2018)



### 3.3 Model quality estimation

Besides the qualitative discussion above, we estimated the quality of our predicted peatland extent through two different approaches. First, we compared our model results against the training data detailed in Section 2.3. For this analysis, we performed a BLOO CV as described in Section 2.4.2. Peat-ML(CV) accuracy was an $r^2$ of 0.72, a MBE of -0.29%, and a RMSE of 9.11% (Fig. 9b). The model performance across each of the twelve training blocks can be seen in Figs. A3 and A4. While the mean $r^2$ across all training blocks was 0.72, it ranged from a low of 0.20 (predicting for block F in the BLOO CV in Fig. 2) to a

high of 0.88 (block E). One caveat of our error estimation presented here is that we are computing it based upon the datasets used for model training. If these datasets themselves have errors or omissions, as would be expected, then this will diminish the accuracy of our error estimation, as well as the quality of the ML model itself, since they form the benchmark Peat-ML is compared against.

Peat-ML likely underestimates peatland coverage as can be seen from its negative MBE (also visible in the regression

line shown in Fig. 9). We hypothesize this low bias may stem from the use of biomes/ecoregions to denote peatland-free areas. It is possible, since these regions are fairly coarsely defined, that we may be inadvertently assigning small-scale, niche peatland areas as non-peatlands (although we take measures to avoid this, see Section 2.3). If that is the case, we would be training the model to miss the characteristics of these more niche peatland environments and hence biasing our results. We use the ecoregions/biomes from Olson et al. (2001) to delineate these non-peatland regions to counter the fact that high-quality

peatland datasets are typically created only for peatland-rich regions. Without inclusion of this peatland-poor training data, we would be providing the algorithm only peatland-rich training data leaving the model poorly trained for peatland-poor regions. Machine learning algorithms are best suited to interpolation problems (e.g., McCartney et al., 2020) so it is best to produce training data that gives the full range of conditions under which the model is expected to produce predictions. Additionally, for the peatlands of South America, we found that we were over-predicting peatland extents as determined by expert opinion

and field observation, primarily due to the paucity of high-quality peatland maps from the continent. As more high quality peatland mapping products become available from presently poorly mapped regions, the use of these ecoregions/biomes could be removed or reduced.

The second approach to estimate the quality of our peatland map focuses on the Boreal Plains (BP) region of Canada where we have several peatland products for comparison (Fig. A8). The DUC remote-sensing based dataset for this region is uniquely

well ground truthed with over 5000 site visits over its 74.1 $\times 10^4$ km$^2$ area. The DUC dataset has a peatland extent of 186 $\times 10^3$ km$^2$ (Table 3) for the BP region, which is about the same as PEATMAP (whose underlying data source in this region is Tarnocai et al. (2011)). Peat-ML (CV) estimates 199 $\times 10^3$ km$^2$(this is derived from the BLOO CV simulations to allow a more fair comparison, it is 185 $\times 10^3$ km$^2$ when estimated by the full model; Peat-ML) while Hugelius et al. (2020) estimates 164 $\times 10^3$ km$^2$ and Webster et al. (2018) estimates 269 $\times 10^3$ km$^2$. We can estimate a confidence interval using $\pm 2\times$ the Peat-ML (CV)

RMSE which gives a range of 140 $\times 10^3$ to 234 $\times 10^3$ km$^2$. This range suggests that the predicted extent is only significantly different between Peat-ML(CV) and Webster et al. (2018). Given its quality, we take the DUC dataset as our benchmark and use it determine the accuracy of Peat-ML and other products (Table 4). As expected Peat-ML compares well with the DUC



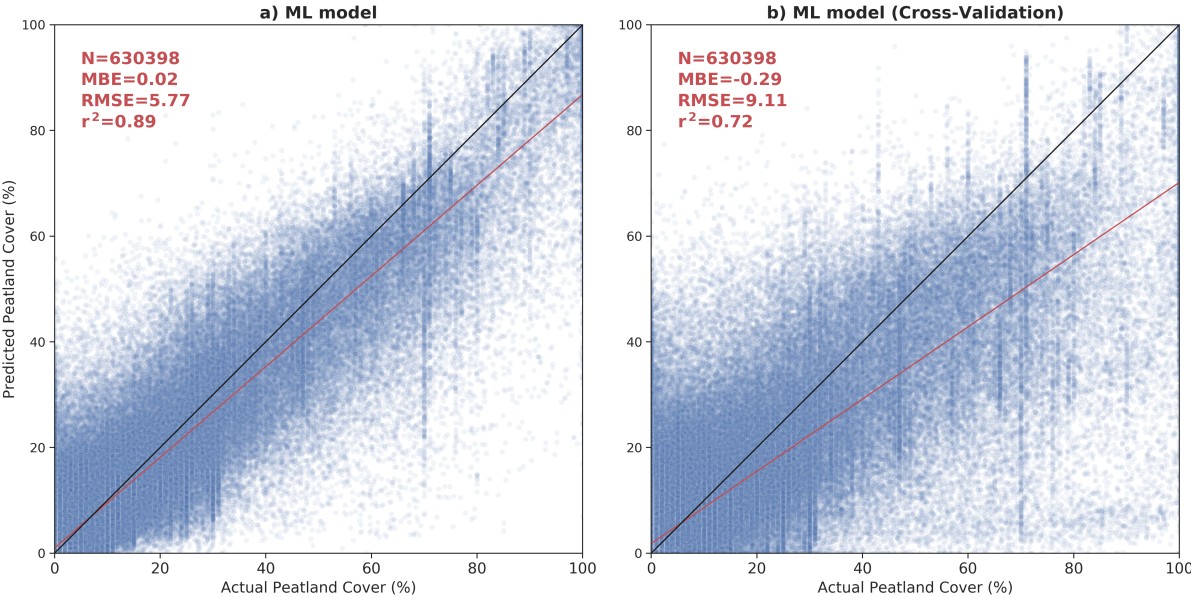

**Figure 9.** Scatter plots of Peat-ML predicted versus actual peatland cover (from the datasets listed in Table 2) for the full model (left panel) and as determined by the BLOO CV (right).

dataset, as it is trained using that dataset. A more useful comparison is with Peat-ML (CV), where the model is not trained with the DUC dataset. Peat-ML (CV) has the second lowest RMSE, mean bias, and explained variance scores after Tarnocai et al.
(2011) in all instances (Table 4). For the DUC region, the Peat-ML (CV) results indicate a higher predictive performance than a peatland mapping product based on soil databases (Hugelius et al., 2020), another based on boosted regression trees using forest structure maps, bioclimatic variables, and surface slopes (Webster et al., 2018), and one based upon ML models informed by expert assessment, although BAWLD has the lowest spatial resolution which may have impeded its performance against the high-resolution DUC dataset. Peat-ML (CV) is, however, outperformed by a more traditional and labour-intensive product
based on air-photo interpretation and soil surveys (Tarnocai et al., 2011), although the performance difference is relatively small (e.g. RMSE difference of 0.39%). This indicates that our model, for this region at least, is of similar or higher quality to other peatland mapping products available from a diverse range of methodologies.

## 3.4  Limitations of our approach

The purpose of our study is to produce a map of peatland distribution for use as an input geophysical field for ESMs with
integrated peatland models. It is tempting to ask whether our technique can give any insights into peat formation or the conditions necessary for a peatland to develop and persist. While our approach is not prescriptive like Hugelius et al. (2020), where peatlands are defined based upon the soil carbon at a location, it is challenging to derive causal information from our simulations. Many of the top features determined by the LightGBM algorithm (Fig. 3) are related to geomorphological characteristics, soil carbon, vegetation and soil water status, and climate. However, peatlands themselves will alter the environment they form



**Table 4.** Statistical comparison of peatland map products as evaluated against the DUC dataset (Smith et al., 2007). RMSE is the root mean squared error. The explained variance score (calculated as $1 - \frac{\sigma^2(y-\hat{y})}{\sigma^2(y)}$, where $y$ is the observations, $\hat{y}$ is the prediction, and $\sigma$ is the standard deviation) has a best possible value of 1.0 with lower scores indicating worse performance.

| Mapping product | RMSE (%) | Mean bias (%) | Explained variance score (-) |
|---|---|---|---|
| Peat-ML | 12.60 | 0.18 | 0.68 |
| Peat-ML (CV) | 17.50 | -1.52 | 0.38 |
| Hugelius et al. (2020) | 18.00 | 2.61 | 0.35 |
| PEATMAP [4] | 17.11 | -0.06 | 0.40 |
| Webster et al. (2018) | 23.25 | -9.49 | 0.07 |
| BAWLD Olefeldt et al. (2021) | 22.24 | -9.33 | 0.16 |

[4] Tarnocai et al. (2011) is the underlying data source for PEATMAP in the DUC domain

within (e.g. fill in depressions with peat, alter the hydrologic balance for the vegetation) and thus it is difficult to differentiate cause from effect.

A weakness of our approach lies in the availability of training data. Our training data for peatland distribution is generally biased towards the high latitudes. While we have good coverage of peatland presence in Canada and West Siberia (see Section 2.2), we presently lack extensive high quality peatland distribution maps for much of the Southern Hemisphere and tropics. Although we expect new products to become available through time (e.g., Anda et al., 2021; Bourgeau-Chavez et al., 2021). As one of our main predictors is sensitive to vegetation (SWIR3), there is also the possibility that peatland types that are not represented in our training data (e.g. mangroves and marshes in Neotropics or papyrus marshes of Africa), will be poorly represented by the available training data that the ML algorithm uses to derive a relationship between the vegetation-based predictor and peatland extent. An additional challenge is the importance of seasonality of covariates (e.g. climate, vegetation indices) that differ significantly between the tropics and high latitudes based on their local dynamics. This may be addressed in future versions of Peat-ML by training separate models for both regions alongside predictors tailored to the dynamics of each region, although that depends on a greatly increasing availability of tropical training datasets to ensure well-trained models.

Also, as discussed in Section 3.3, it would be beneficial to include mapping products for regions where peatlands are relatively sparse. As our peatland sampling strategy was determined by the availability of high-quality peatland maps, we were not able to choose systematic (Rocha et al., 2021) or feature-based sampling strategies that could be more optimal for peatland prediction. Our approach would also benefit from greater availability of processed, global-scale products that should be sensitive to water status below the peat surface like L-band SAR (e.g. Touzi et al., 2013).





# 4 Conclusions

We present a new global peatland fractional coverage map, Peat-ML, at a scale of 5 arc minutes resolution. Peat-ML was
generated using machine learning techniques drawing upon drivers of peatland formation which include spatially distributed
climate, soil, geomorphological, and vegetation data. The ML model was trained using maps of peatland fractional coverage
for 14 relatively extensive regions along with masks of non-peatland areas. To evaluate Peat-ML, we qualitatively compared
it to other available peatland maps, and we also quantified model performance in two approaches. The first approach is based
on a blocked leave-one-out cross-validation strategy designed to minimize the influence of spatial autocorrelation. Based
upon that approach, Peat-ML has an average $r^2$ of 0.73 with a root mean squared error and mean bias error of 9.11% and -
0.36%, respectively, evaluated against our model training data. Our second model quality estimate was generated by comparing
Peat-ML against a high-quality, extensively ground truthed map for the 74.1 $\times 10^4$ km$^2$ Canadian Boreal Plains region. This
comparison suggests Peat-ML is of comparable or higher quality than other presently available peatland mapping products.
Future versions of Peat-ML would benefit from further high quality and ground truthed datasets of peatland extent, especially
in tropical regions.

# 5 Code and data availability

Python code for the statistical modelling is available at https://gitlab.com/jormelton/peatlandmachinelearning (After accepted
for final publication it will be archived in a Zenodo repository as well). A netCDF format version of the Peat-ML is available
at https://doi.org/10.5281/zenodo.5794336.

*Author contributions.*  Following the CRediT taxonomy, JRM conceptualized the study. EC, JRM, and MF performed data curation, formal
analysis, investigation, and software and methodology development. JMM-L, RSW, and KM also contributed to methodology development.
KM, JMM-L, HC-Q, and DK provided resources. JRM and EC did the visualization. Validation was done by RSW, JMM-L, HC-Q, LVV,
DK, EC, and JRM. JRM wrote the original draft of the manuscript. All authors reviewed and edited the final manuscript.

*Competing interests.*  The authors declare that they have no competing interests.

*Acknowledgements.*  We acknowledge the efforts of Yuanqiao Wu and Diana Verseghy who lead an earlier effort to predict global peatland
extents using machine learning approaches. We thank Dirk Flugmacher, Matt Aitkenhead, Fokke Brouwer, Freddie Draper, Greta Dargie,
and Rudiyanto for sharing their peatland mapping products. We also thank Camila Delgado-Montes for processing the Rio Madre de Dios
data. We have adopted the colour bar scheme from Hugelius et al. (2020) for our peatland extent plots. Lastly, we thank Michel Bechtold for
comments about an earlier study that we used to improve the design of this study.



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



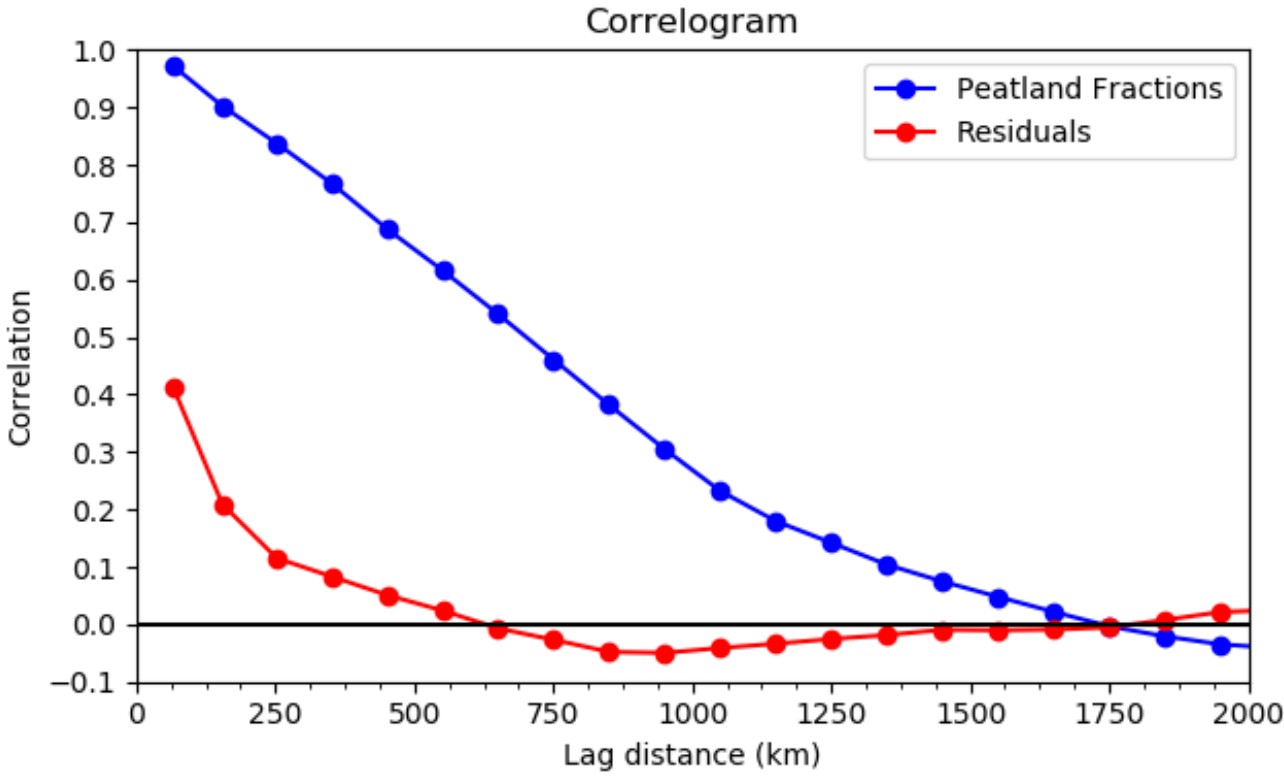

**Figure A1.** Correlelogram showing the spatial correlation between model residuals as a function of distance computed using Moran's I.



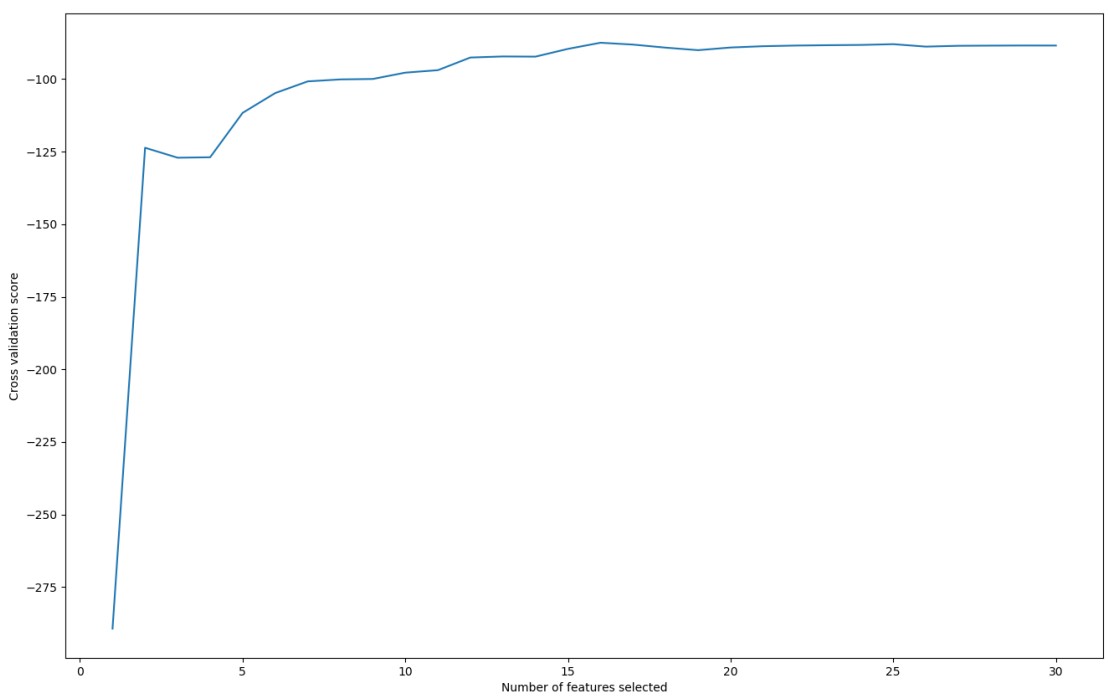

**Figure A2.** Cross validation scores against the number of features selected by RFECV (see Section 2.4.3).





**Figure A3.** Scatter plots of full model (Peat-ML) predicted peatland extent and extent from the peatland training datasets over the 14 BLOO CV blocks.





**Figure A4.** Scatter plots of the CV trials for Peat-ML (Peat-ML CV) predicted peatland extent and extent from the peatland training datasets over the 14 BLOO CV blocks.

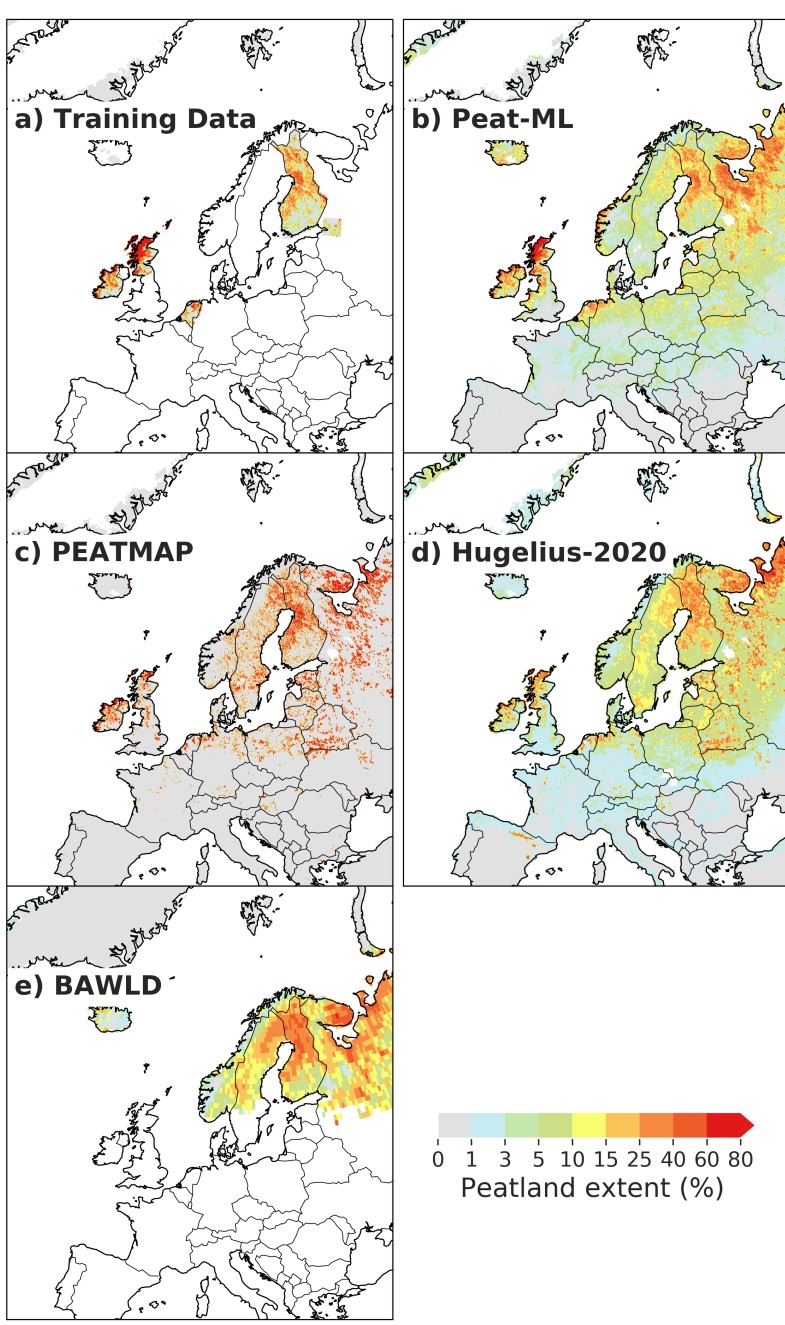

**Figure A5.** Maps of western European peatlands including a) training data used by the ML model, b) Peat-ML predicted peatlands and the peatland coverage from c) PEATMAP (Xu et al., 2018), d) Hugelius et al. (2020), and e) BAWLD (Olefeldt et al., 2021), whose domain only partly extends over the region displayed.

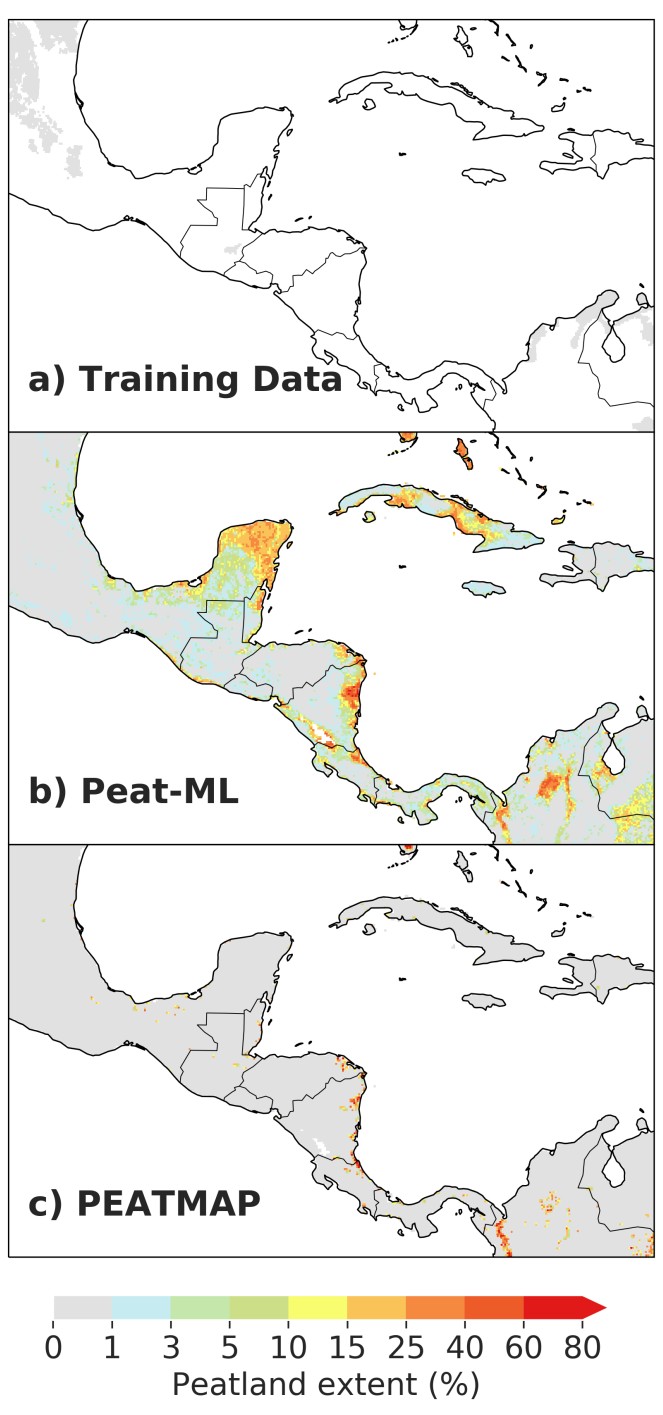

**Figure A6.** Peatland extent over Central America. Panel a) shows the ML training data, b) the Peat-ML predicted peatland extent, and c) the PEATMAP extent from Xu et al. (2018)

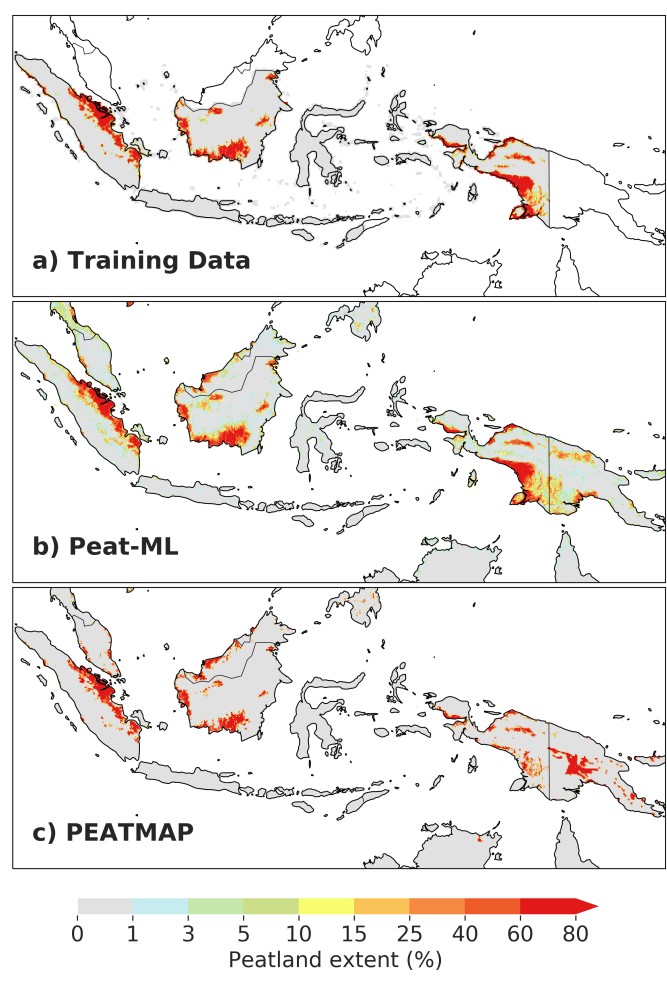

**Figure A7.** Peatland extent over the Indonesian archipelago. Panel a) shows the ML training data, b) the Peat-ML predicted peatland extent, and c) the PEATMAP extent from Xu et al. (2018)





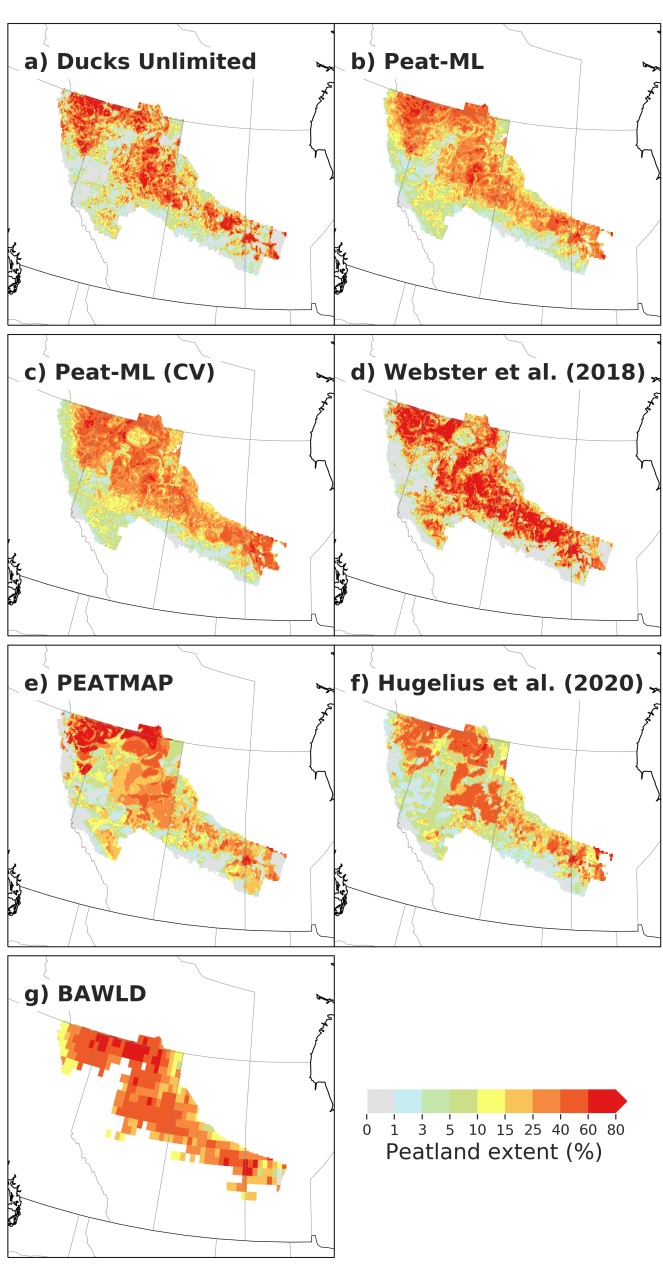

**Figure A8.** Maps of peatland extent for the Boreal Plains of Canada.





**Table A1.** The 30 predictors selected using VIF with a threshold value of 5. The final 16 features further selected by the RFE-CV algorithm for use in the final model are listed in Table 1. See Section 2.2.1 for further discussion on the variable processing.

| Category | Shortname | Variable | Data Source |
|---|---|---|---|
| Climate[1] | soil_DJF | soil water | Terraclimate (Abatzoglou et al., 2018) |
| | srad_DJF | downward surface shortwave radiation | |
| | swe_DJF | snow water equivalent | |
| | ws_DJF | wind speed | |
| | vap_MAM | vapour pressure | |
| | ro_SON | runoff | |
| | pdsi_SON | Palmer Drought Severity Index | |
| Soils | OLM_soil_organic_carbon_30cm | organic carbon content | Open Land Maps (Hengl, 2018) |
| | OLM_Soil_BulkDensity_30cm | soil bulk density | |
| Vegetation | Dormancy_1 | dormancy | MODIS (MCD12Q2 V6) |
| | Senescence_2 | senescence | (Friedl et al., 2019) |
| | EVI_Amplitude_2 | enhanced vegetation index amplitude | |
| | EVI_Area_1 | Sum of EVI1 from Greenup to Dormancy | |
| | EVI_Area_2 | Sum of EVI2 from Greenup to Dormancy | |
| | minMODIS_NPP | minimum NPP | MOD17A3 V055 |
| | | | (Running et al., 2011) |
| | SWIR3_reflectance_mean_SON | Shortwave infrared radiation reflectance [2] | S-NPP VIIRS |
| | | | (Didan and Barreto, 2018) |
| Terrain | spi | stream power index [3] | Geomorpho90m |
| | geom | geomorphon | (Amatulli et al., 2020) |
| | slope | slope | |
| | tcurve | tangential curvature [4] | |
| | rough-scale | scale of terrain roughness | |
| | dev-magnitude, dev-scale | maximum elevation deviation value | |
| | dx | first directional derivative (east-west) [5] | |
| | dxy,dyy | second directional derivative [6] | |
| | convergence | convergence index [7] | |
| | aspect-sine, aspect-cosine | sine(cosine) of aspect [8] | |
| | northness | northness [9] | |

[1] The means of DJF, MAM ,JJA, and SON refer to the 3-month periods indicated by the first letter of each month, respectively.   [2] 2225 - 2275 nm   [3] product between the upstream catchment area and the tangent of the local slope angle   [4] measures the rate of change perpendicular to the slope gradient and is related to the convergence and divergence of flow across a surface   [5] the rate of change of the elevation in a specific direction   [6] the rate of change of the slope in a predetermined direction   [7] terrain variable that details the convergent areas as channels and divergent areas as ridges. It varies from a value of -100 for ridges, 0 for planar or flat areas, and up to 100 for sink areas.   [8] angular direction that a slope faces   [9] calculated from sine of the slope multiplied by the cosine. Northness gives a continuous measure of the orientation combined with the slope. For the northern hemisphere, a northness approaching 1 gives a northern exposure on a vertical slope (that is a slope exposed to very low amount of solar radiation), conversely a northness of -1 gives a very steep southern slope that would be highly exposed solar radiation.





**Table A2.** LightGBM hyperparameters that underwent Bayesian optimization and their final optimized values. See Section 2.2.1 for further discussion on the variable processing. See Pedregosa et al. (2011) documentation for further discussion about each hyperparameter.

| Name | Range | optimized value |
|---|---|---|
| boosting_type | gbdt, dart, goss | dart |
| num_leaves | 10 - 50 | 30 |
| n_estimators | 50 - 300 | 250 |
| learning_rate | 0.005 - 0.4 | 0.18817013045111064 |
| max_bin | 25 - 300 | 95 |
| max_depth | -1 - 15 | 6 |
| subsample_for_bin | 20000 - 300000 | 80000 |
| min_child_samples | 5 - 60 | 10 |
| reg_alpha | 0 - 1 | 0.705705986914311 |
| reg_lambda | 0 - 1 | 0.9086692536858783 |
| colsample_bytree | 0.5 - 1.0 | 0.8251441062858274 |