# Peer review of "A map of global peatland extent created using machine learning (Peat-ML)"

_Geoscientific Model Development, 2021_

## Author Comment (AC2)

**Dear Editor,**

Please find below our reply to Reviewers #1 (Dr. Julie Loisel) and #2. We copy their reviews in full and then provide our replies in **bold font**.

**Reviewer #1 (Julie Loisel)**

The authors provide a peatland map that was realized using a machine-learning approach. Data that were used to inform the mapping exercise include climate, soil, and remotely-sensed vegetation, and geomorphology/terrain. The model was trained using existing areas of peatland vs. non-peatland that were previously mapped by others. I very much enjoyed reading the manuscript. The literature background is well informed and the methods are described with sufficient details, in addition of being sound. I must say that I am not an expert on machine learning, so I cannot comment on this aspect of the study. An important result is the list of predictive features for peatland presence; I am particularly glad to see geomorphological landform and terrain slope as the most important ones. It reassures me to see these results, as they make intuitive sense and relate well with field observations, even though these localized landscape features may be challenging to represent in a model. The regional map products are well described and contrasted with existing literature. Likewise, the qualitative discussion on regional results is sound; in particular, I appreciate the authors' aknowledgement of limited ground-truthing data across the tropics as well as the dynamic ecohydrological processes operating across the tropics, which may limit confidence in their results and/or the power of their ML approach. LAstly, the model limitations are clearly spelled out.

I recommend publication of this manuscript as is.

Reviewer: Julie Loisel

We thank Dr. Loisel for the review and positive comments. We are glad to hear that our work was seen as potentially valuable.

**Reviewer #2**

In this manuscript, Melton et al. employed a machine learning approach, i.e., Gradient Boosting Decision Tree (LightGBM) to predict the coverage of peatland over the whole globe by using climate, soil, terrain and vegetation data. The training data for LightBGM were from high-resolution regional dataset of peatland distribution and coverage. It is found that their approach can map peatland coverage reasonably well compared with existing global products using traditional approaches. As stated by the authors, the results of the study are valuable for prescribing peatland cover fraction in the Earth System models to better explore the impact of climate changes on global or regional carbon and water budgets.

In general, the manuscript is well-written. The use of approach and data are appropriate. The results seem to be robust. There are, however, several weaknesses of the analyses, which require further improvement before being accepted by GMD.

**Thank you for your comments and the time taken for the comprehensive review**

Major comments:

- 1. Issues with the machine learning approach:
- 1.1. Lack of a baseline model to illustrate the added value of using LightGBM.

The authors have had quite detailed explanations of their machine learning workflow, but the performance of a baseline model is missing, which makes it difficult to judge if the choice of LightGBM has helped to improve the prediction or not.

We did investigate alternative models (can be seen in our code, see the Zenodo repository or the Gitlab link), however since the purpose of this paper was to produce a skillful map of global peatland distribution, we only presented the results of the model we found to be most advantageous in terms of model skill and computational cost. We could present the results of a simple model like linear regression (an option in our code), but that would only demonstrate that our model outperforms an extremely simple model, making its inclusion then of dubious value to the paper especially considering the primary purpose of our work. Through the course of our project, other ML algorithms were investigated such as Random Forests (RF), however RF is much slower than LightGBM and we noticed no significant change in model skill. LightGBM's speed also allowed us to use more computationally expensive techniques such as the removal of highly multicollinear features iteratively via VIF. Lastly, it is also unclear to us how a nontrivial baseline model would be best chosen without it leading to excess additional simulations given the numerous potential choices of baseline models available, e.g. RF, Support Vector Machine, Convolutional Neural Network, Adaboost, Multilayer Perceptron, Naive Bayes, etc. Based upon these points, we feel that our choice to not include a baseline model is inline with the purpose of our paper and its aims and the inclusion of one would not improve our manuscript significantly.

1.2. The training data shown in Fig. 2 seem not to be well-balanced for producing meaningful machine learning outcomes. To me, there seem to be too many grids with either high peatland cover fraction (>60%) or zero peatland cover, while the grids with medium peat cover fraction (e.g., 1-25%) are rare. I strongly suggest the authors plot a figure showing the probability of peatland cover fraction in the whole training dataset to see if the training dataset is balanced or not. If not, the authors need to address the problem with an appropriate method. For instance, the use of non-peatland grids needs to be constrained to maintain a balanced training dataset.

This is an interesting suggestion. First, we note that 'balance' of a dataset is an important consideration in the context of classification problems, not regression problems (as we have here). Regardless, a representative training dataset is the underlying consideration suggested and we should reflect on what would truly represent a 'well-balanced' training set. Should a balanced training set contain roughly equal numbers of grid cells from 0 to 100%, i.e. a flat histogram, or should a balanced training dataset look like the 'true' distribution of peatlands? Globally peatlands cover roughly 3% of the land surface (Frolking et al. 2011), which would suggest a highly skewed histogram. We favour the latter. A training distribution more like the true global proportions, rather than a uniform one, would likely lead to better peatland extent predictions as the model would be better trained on the combination of peatland predictors that lead both to peatland existence and its lack thereof. Based upon that, arguably a training dataset then should contain some peatland coverage in roughly 3% of its training cells to best represent the global distribution of peatlands on the Earth's surface. Our present peatland training dataset has ~30% of its grid

cells with non-zero peatland coverage. Although the 3% number should have quite large uncertainty bounds, our training dataset is likely too peatland rich. This would indicate that for 'balance' it may be beneficial to include more non-peatlands (provided such data was available - and we are not aware of any).

To further look at our training distribution, the figure below shows a histogram of our present distribution of training and model predicted peatland extent. The histogram suggests that PeatML reasonably reproduces the training distribution. The training distribution is weighted toward no peatland coverage, as mentioned previously, but then has relatively uniform (but decreasing) distribution up until about 70% then with a small increase for 100% coverage. It seems intuitively reasonable that 100% peatland coverage on a 5 arc minute grid would be more uncommon due to sub-grid heterogeneity of the land surface. PeatML generally has a similar distribution to its training data, but does have a general underprediction, as noted in our paper.

1.3. The training data used in the study (Table 2) are from existing regional peatland mapping data generated from various approaches. The training data itself has various biases and is not ground truth. I am wondering why the authors did not choose to use the available ground-truthed site-level observations data for their machine learning model? Is such data not sufficient enough to make global predictions?

For our training datasets, we purposefully selected datasets that had performed some form of skill evaluation or ground truthing (described for each dataset in Section 2.3), but we fully agree the training datasets contain biases (which we discuss on line 471). This is not unique to our training dataset, rather it is a commonality of all large-scale observational datasets. More specifically, regarding the use of site-level observations, our training dataset includes over 630,000 pixels with peatland fractional coverage (see Fig. 9). If we were to try and include site-level peatland estimates, or use them solely, this number would likely amount to only a few thousand data points. This would hamper our ability to effectively train the model not only due to the small amount of training data, but it is also unclear how that would affect the training distribution as discussed above. There is thus a trade-off to be had. Either one chooses to use large-scale peatland mapping products - with biases (both quantified and not) - and therefore is able to better train the machine learning model or uses site-level observations with little bias but with a resulting poorly

**constrained ML model. We feel that the former option is most appropriate for global scale peatland estimation.**

1. The data used to train and evaluate the machine learning model are not comprehensive and state-of-theart. The authors have made a good effort in bringing available peatland cover data all over the world to train the model. But I was a bit disappointed that they have not used any recent dataset from China (e.g., Mao et al. 2020) or the US, or at least explain why they have not done so. Given the lack of training and evaluation dataset as also pointed out by the authors in the manuscript, I suggest they take advantage of the available dataset as much as possible.

We agree that we likely don't have every potentially relevant peatland mapping dataset included in our training dataset. As discussed above, and in our paper, we sought out high quality, relatively large-scale, validated, and readily available peatland mapping products. Based upon those criteria we included all datasets that we were aware of, but there may be more datasets published in more obscure journals that we missed. Thank you for noting the Mao et al. 2020 paper, we were unaware of it previously. However, it is not a peatland specific mapping product (none of the wetland classes it categorizes are peatlands) and hence it is not suitable to be used for model training. As to the US datasets alluded to, no examples were given so we are not sure which studies were intended here. Regardless, we do intend to integrate new mapping datasets into future versions of PeatML as they become available, especially for poorly mapped regions like Africa and South America.

2. The choice of the spatial resolution of the input dataset (5x5 arc minute grid) for machine learning is a bit arbitrary to me. What is the rationale behind the choice? Why not use coarser spatial resolution or higher spatial resolution? Will the choice of the spatial resolution affect the machine learning results?

Yes, it is, naturally, somewhat arbitrary. We chose a resolution that would be reasonably close to the covariates (predictors) datasets original spatial resolution. Our chosen spatial resolution allows the scale of the interpolation to a common grid to be as minimized as possible. Changing the spatial resolution should have some impact upon the model results. For example, if the spatial resolution was instead 0.5 degrees, many of the 5 arc minute pixels would be averaged within a 0.5 degree grid cell. For a region with relatively few high coverage peatland areas and little peatland extent elsewhere, that distribution would be lost and the model would be trained to produce some mean value at the 0.5 degree scale. Similarly, if the spatial resolution is too high, the model may be predicting peatland extent using co-variates that are of coarser spatial extent than the chosen grid cell resolution, thus confounding the model's attempts to associate the peatlands with their co-variates. Also there is a computational cost with increasing number of gridcells thus while 5 arc minutes is a much higher spatial resolution than our final intended use in a land surface model (0.5 - 1 degree) it is also not so high as to be computationally impractical.

1. The authors have done a systematic job in selecting the best predictors for the machine learning model (Figure 3 and Table 1). However, the selected predictors are a mixture of climate, soils, vegetation and terrain variables. Some are relatively stable through time (e.g., terrain and soil), some are not (e.g., climate and vegetation). I understand that such a combination gives the best global prediction. But I am also curious how the machine learning model performs with only terrain and soil variables, or with only climate and vegetation variables. This may provide some useful insights on the relative importance of long-lived vs short-lived conditions in forming peatland. As we state in Section 3.4 of our paper, the principle purpose of our research was to produce a map of peatland distribution for use as an input geophysical field for LSM/ESM simulations. It could be an interesting follow-on to attempt to elucidate more mechanistic information based upon the predictors, but we don't feel that this is a necessary inclusion to the present work.

Other comments:

Line 5-6: please be more specific about which "machine learning technique" used in this study.

**For the purpose of an abstract, we prefer to keep the focus on the model results and not the specifics of the algorithms, especially as we use several techniques (see Fig. 1).**

Line 79-81: have the authors tested if using different definitions of peatland instead of "30% dead organic material by dry mass" and "minimum thickness of 30 cm of peat", will affect the results?

**No, we haven't tested others. Since the definition of peatlands is, due to the nature of these ecosystems, arbitrary and only comes from community consensus, we simply chose what we perceived to be the most commonly accepted definitions in the literature.**

Line 86-87: Why use 5x5 arc minute grid? Please explain.

**Discussed above**

Line 97-98: please explain here why use "calculated length of the longest day of the year (hours) for each cell" as a predictor?

**The idea behind this predictor was that it could be used as a means to delineate tropical vs. extratropical regions. We have added the following text, 'The longest day of year could be used by the model to determine tropical versus extratropical regions'.**

Line 103: please specify the years covered by JRA-55.

**According to the JRA-55 original publication, it runs from 1958 - 2013. JRA-55 was used to help generate Terraclimate, we didn't use it by itself.**

Line 115: it is ambiguous to say "we use 30 cm depth estimate for all soil variables". Do you use 0-30 cm averaged value or the value only at 30 cm depth?

**The SoilGrids dataset provides 0, 30 cm, etc. depth intervals. As it is a machine learning product, the 30 cm depth is the model estimate at that depth. We believe our present wording is thus correct.**

Line 133-134: please be clearer if you use the multi-year average of "2000-2015 period" or the data from each year?

**We changed the wording from 'We used the available 2000 – 2015 period.' to 'We averaged the data over the available 2000 – 2015 period.'**

Line 143: what are the years of S-NPP dataset used in this study?

**2012 - 2019**

Line 191-192, 201-202: The transformation of peatland map into cover fraction seems to be arbitrary. Please explain why?

It isn't arbitrary but rather follows our stated definition of peatlands. Our definition of a peatland, as discussed above and in our paper, is based on the existence of >30 cm of peat, thus any pixel in the peatland mapping dataset with greater than 30 cm of peat is peatland. If a pixel has less peat thickness, following the definition, it isn't peatland.

Line 235-244: The creation of zero peatland cover grids is also a bit arbitrary to me. Generating too many zero grids in the training dataset, might give rise to an imbalanced dataset that is detrimental to the performance of the machine learning model.

**Discussed above**

Line 284: Please specify how Rj is derived.

**It is computed from the vector of correlations calculated between the independent variables and the dependent variables and the correlation matrix of correlations between the independent variables. We can add the relevant equations to our revised manuscript.**

Line 337-343 and Figure 3: why the importance (%) of wind speed and runoff from the full model simulation do not match with that from the BLOO CV, which is different from the other variable? The explanation provided in the text is not clear to me.

The fitted model(s) between the BLOO CV models and the final model are not the same and thus the predictor importance can vary. The BLOO CV fits a model over all training data but that within a specified CV block. That means there are 14 different models with their own rankings of feature importance. Each of those models gets a slightly different feature importance ranking because it is exposed to slightly different training data (as it excludes, by design, roughly one-fourteenth of the training data). The full model, conversely, sees all training data. Because of these differences in the training data used to fit each model, it is expected that feature importance could vary.

Line 353: Is the lower peatland cover over the northern hemisphere than the other available estimate related to the imbalance of the training dataset (too many zero peatland cover grids in the training dataset)?

**See above**

Table 1 and 2: please add original spatial resolution of each dataset in the table. Also, it would be great to include the time coverage of each dataset, and how they were aggregated timely.

**Added**

Figure 2: It would be nice to add a plot showing the probability of different peat cover fraction in the training dataset.

**We don't understand the suggestion. The plot at present shows the peatland percent cover of the training dataset. It is not clear what probability is meant here.**

Figure 6: We need more quantitative comparison among different dataset. In addition, I am not quite sure how "Peat-ML from the BLOO CV" is derived in this figure. Did you average the results from all the 14 BLOO CV, or do you use only the results from which the blocks covering Canada were left out. In the figure caption, "three other peatland extent products (d-f)" should be "four other peatland extent products (d-g)".

Fig. 6 shows PeatML compared to its training data, four other peatland mapping products, along with the peatland map derived from the BLOO CV. We purposefully chose to keep the comparison between these different datasets qualitative. None of these datasets should be considered 'correct' and thus a more quantitative comparison would be unwarranted or give the impression of higher certainty in the peatland predictions of these datasets. We reserve quantification of model error to the BLOO CV and the comparison against the Ducks Unlimited data which we argue is of sufficiently high quality to treat as such due to its exhaustive ground truthing. For Figure 6 the BLOO CV result is the LightGBM model predicted peatland extent when that block was excluded from the BLOO CV, thus this is the model predicting upon a region that it did not train upon. Thanks also for noticing the mislabelled number, we have corrected it to four.

References:

Mao, D.H., Wang, Z.M.\*, Du, B.J., Li, L., Tian, Y.L., Zeng, Y., Song, K.S., Jiang, M., Wang, Y.Q. 2020. National wetland mapping in China: A new product resulting from object based and hierarchical classification of Landsat 8 OLI images. ISPRS Journal of Photogrammetry and Remote Sensing, 164: 11-25

Citation: https://doi.org/10.5194/gmd-2021-426-RC2

Frolking, S., Talbot, J., Jones, M. C., Treat, C. C., Kauffman, J. B., Tuittila, E.-S., and Roulet, N.: Peatlands in the Earth's 21st cen-tury climate system, Environ. Rev., 19, 371–396, 2011